# WHEN PREDICT CAN ALSO EXPLAIN: FEW-SHOT PREDICTION TO SELECT BETTER NEURAL LATENTS

## ABSTRACT

Latent variable models serve as powerful tools to infer underlying dynamics from observed neural activity. Ideally, the inferred dynamics should align with true ones. However, due to the absence of ground truth data, prediction benchmarks are often employed as proxies. One widely-used method is *co-smoothing*, which involves jointly estimating latent variables and predicting observations along held-out channels to assess model performance. In this study, we reveal the limitations of the co-smoothing prediction framework and propose a remedy. Utilizing a student-teacher setup with Hidden Markov Models, we demonstrate that the high co-smoothing model space encompasses models with arbitrary extraneous dynamics within their latent representations. To address this, we introduce a secondary metric— *few-shot co-smoothing*. This involves performing regression from the latent variables to held-out channels in the data using fewer trials. Our results indicate that among models with near-optimal co-smoothing, those with extraneous dynamics underperform in the few-shot co-smoothing compared to 'minimal' models that are devoid of such dynamics. We also provide analytical insights into the origin of this phenomenon. We further validate our findings on real neural data using two state-of-the-art methods: LFADS and STNDT. In the absence of ground truth, we suggest a novel measure to validate our approach. By cross-decoding the latent variables of all model pairs with high co-smoothing, we identify models with minimal extraneous dynamics. We find a correlation between few-shot co-smoothing performance and this new measure. In summary, we present a novel prediction metric designed to yield latent variables that more accurately reflect the ground truth, offering a significant improvement for latent dynamics inference. Code available here.

## 1 INTRODUCTION

In neuroscience, we often have access to simultaneously recorded neurons during certain behaviors. These observations, denoted $X$, offer a window onto the actual hidden (or latent) dynamics of the relevant brain circuit, denoted $Z$ (Vyas et al., 2020). Although, in general, these dynamics can be complex and high-dimensional, capturing them in a concrete mathematical model opens doors to reverse-engineering, revealing simpler explanations and insights (Barak, 2017; Sussillo & Barak, 2013). Inferring a model of the $Z$ variables, also known as latent variable modeling (LVM), is part of the larger field of system identification with applications in many areas outside of neuroscience, such as fluid dynamics (Vinuesa & Brunton, 2022) and finance (Bauwens & Veredas, 2004).

Because we don't have ground truth for $Z$, prediction metrics on held-out parts of $x$ are commonly used as a proxy (Pei et al., 2021). However, it has been noted that prediction and explanation are often distinct endeavors (Shmueli, 2010). For instance, Versteeg et al. (2023) use an example where ground truth is available to show how different models that all achieve good prediction nevertheless have varied latents that can differ from the ground truth. Such behavior might be expected when using highly expressive models with large latent spaces. Bad prediction with good latents is demonstrated by Koppe et al. (2019) for the case of chaotic dynamics.

Various regularisation methods on the latents have been suggested to improve the similarity of $Z$ to the ground truth, such as recurrence and priors on external inputs (Pandarinath et al., 2018), low-dimensionality of trajectories (Sedler et al., 2022), low-rank connectivity (Valente et al., 2022; Pals

et al., 2024), injectivity constraints from latent to predictions (Versteeg et al., 2023), low-tangling (Perkins et al., 2023), and piecewise-linear dynamics (Koppe et al., 2019). However, the field lacks a quantitative, *prediction-based* metric that credits the simplicity of the latent representation—an aspect essential for interpretability and ultimately scientific discovery, while still enabling comparisons across a wide range of LVM architectures.

Here, we characterize the diversity of model latents achieving high *co-smoothing*, a standard prediction-based framework for Neural LVMs, and demonstrate potential pitfalls of this framework. We propose a few-shot variant of co-smoothing which, when used in conjunction with co-smoothing, differentiates varying latents. We verify this approach both on synthetic toy problems and state-of-the-art methods on neural data, providing an analytical explanation of why it works in a simple setting.

## 2 RELATED WORK

Our work builds on recent developments in Neural LVMs for the discovery of latent structure in noisy neural data on single trials. We refer the reader to Pei et al. (2021) supplementary table 3 for a comprehensive list of Neural LVMs published from 2008-2021. Central to our work is the co-smoothing procedure, which evaluates models based on the prediction of activity from held-out neurons provided held-in neuron activity from the same trial. Co-smoothing was first introduced in Yu et al. (2008) and Macke et al. (2011) for the validation of GPFA as a Neural LVM.

Pei et al. (2021) curated four datasets of neural activity recorded from behaving monkeys and established a framework to evaluate co-smoothing among other prediction-based metrics on several models in the form of a standardized benchmark and competition.

In contrast to prediction approaches, a parallel line of work focuses on explaining and validating Neural LVMs on synthetic data, enabling direct comparison with the ground truth (Sedler et al., 2022; Brenner et al., 2022; Durstewitz et al., 2023). Versteeg et al. (2023) validated their method with both ground truth and neural data, demonstrating high predictive performance with low-dimensional latents.

A concept we introduce is cross-decoding across a population of models to find the most parsimonious representation. Several works compare representations of large model populations (Maheswaranathan et al., 2019; Morcos et al., 2018). They apply Canonical Correlation Analysis (CCA), a symmetric measure of representational similarity, whereas we use regression, which is not symmetric. The application to Neural LVMs may be novel.

One related approach comparing representations in deep neural networks is *stitching* components of separately trained (and subsequently frozen) models into a composite model using a simple linear layer (Lenc & Vedaldi, 2015; Bansal et al., 2021).

Central to our work is the concept of few-shot learning a decoder from a frozen intermediate representation. Sorscher et al. (2022) developed a theory of geometric properties of representations that enables few-shot generalization to novel classes. They identified the geometric properties that determine a signal-to-noise ratio for classification, which dictates few-shot performance. While this setting differs from ours, links between our works are a topic for future research. To our knowledge, the use of few-shot generalization as a means to identify interpretable latent representations, particularly for Neural LVMs, is a novel idea.

## 3 CO-SMOOTHING: A CROSS-VALIDATION FRAMEWORK

Let $\boldsymbol{X} \in \mathbb{Z}_{\geq 0}^{T \times N}$ be spiking neural activity of $N$ channels recorded over a finite window of time, i.e., a *trial*, and subsequently quantised into $T$ time-bins. $X_{t,n}$ represents the number of spikes in channel $n$ during time-bin $t$. The dataset $\mathcal{X} := \{\boldsymbol{X}^{(i)}\}_{i=1}^{S}$, partitioned as $\mathcal{X}^{\text{train}}$ and $\mathcal{X}^{\text{test}}$, consists of $S$ trials of the experiment. The latent-variable model (LVM) approach posits that each time-point in the data $\boldsymbol{X}_{t,:}^{(i)}$ is a noisy measurement of a latent state $\boldsymbol{Z}_{t,:}^{(i)}$.

To infer the latent trajectory $\boldsymbol{Z}$ is to learn a mapping $f : \boldsymbol{X} \mapsto \boldsymbol{Z}$. On what basis do we validate the inferred $\boldsymbol{Z}$? We have no ground truth on $\boldsymbol{Z}$, so instead we test the ability of $\boldsymbol{Z}$ to predict unseen or

held-out data. Data may be held-out in time, e.g., predicting future data points from the past, or in space, e.g., predicting neural activities of one set of neurons (or channels) based on those of another set. The latter is called co-smoothing (Pei et al., 2021).

The set of $N$ available channels is partitioned into two: $N^{\text{in}}$ held-in channels and $N^{\text{out}}$ held-out channels. The $S$ trials are partitioned into train and test. During training, both channel partitions are available to the model and during test, only the held-in partition is available. During evaluation, the model must generate the $T \times N^{\text{out}}$ rate-predictions $R_{:,\text{out}}$ for the held-out partition. This framework is visualised in Fig. 1A.

Importantly, the encoding-step or inference of the latents is done using a full time-window, i.e., analogous to *smoothing* in control-theoretic literature, whereas the decoding step, mapping the latents to predictions of the data is done on individual time-steps:

$$\boldsymbol{Z}_{t,:} = f(\boldsymbol{X}_{:,\text{in}}; t) \tag{1}$$
$$R_{t,\text{out}} = g(\boldsymbol{Z}_{t,:}), \tag{2}$$

where the subscripts 'in' and 'out' denote partitions of the neurons. During evaluation, the held-out data from test trials $\boldsymbol{X}_{:,\text{out}}$ is compared to the rate-predictions $\boldsymbol{R}_{:,\text{out}}$ from the model using the co-smoothing metric $\mathcal{Q}$ defined as the normalised log-likelihood, given by:

$$Q(R_{t,n}, X_{t,n}) := \frac{1}{\mu_n \log 2}\bigg( \mathcal{L}(R_{t,n}; X_{t,n}) - \mathcal{L}(\bar{r}_n; X_{t,n}) \bigg) \tag{3}$$

$$\mathcal{Q}^{\text{test}} := \sum_{n \in \text{held-out}} \sum_{i \in \text{test}} \sum_{t=1}^{T} Q(R_{t,n}^{(i)}, X_{t,n}^{(i)}), \tag{4}$$

where $\mathcal{L}$ is poisson log-likelihood, $\bar{r}_n = \frac{1}{TS}\sum_i \sum_t X_{t,n}^{(i)}$ is a the mean rate for channel $n$, and $\mu_n := \sum_i \sum_t X_{t,n}^{(i)}$ is the total number of spikes, following Pei et al. (2021).

Thus, the inference of LVM parameters is performed through the optimization:

$$f^*, g^* = \text{argmax}_{f,g} \mathcal{Q}^{\text{train}} \tag{5}$$

using $\mathcal{X}^{\text{train}}$, without access to the test trials from $\mathcal{X}^{\text{test}}$. For claritry, apart from equation 5, we report only $\mathcal{Q}^{\text{test}}$, omitting the superscript.

## 4 GOOD CO-SMOOTHING DOES NOT GUARANTEE CORRECT LATENTS

It is common to assume that being able to predict held-out parts of $\boldsymbol{X}$ will guarantee that the inferred latent aligns with the true one (Macke et al., 2011; Pei et al., 2021; Wu et al., 2018; Meghanath et al., 2023; Keshtkaran et al., 2022; Keeley et al., 2020; Le & Shlizerman, 2022; She & Wu, 2020; Wu et al., 2017; Zhao & Park, 2017; Schimel et al., 2022; Mullen et al., 2024; Gokcen et al., 2022; Yu et al., 2008; Perkins et al., 2023). To test this assumption, we use a student-teacher scenario where we know the ground truth. To compare how two models $(u, v)$ align, we infer the latents of both from $\mathcal{X}^{\text{test}}$, then do a regression from latents of $u$ to $v$. The regression error is denoted $\mathcal{D}_{u \to v}$ (i.e. $\mathcal{D}_{\text{T} \to \text{S}}$ for teacher to student decoding). Contrary to the above assumption, we hypothesize that good prediction guarantees that the true latents are contained within the inferred ones (low $\mathcal{D}_{\text{S} \to \text{T}}$), but not vice versa (Fig. 1C). It is possible that the inferred latents possess additional features, unexplained by the true latents (high $\mathcal{D}_{\text{T} \to \text{S}}$).

To verify this hypothesis, we choose both student and teacher to be a discrete-space, discrete-time Hidden Markov Model (HMM). As a teacher model, they simulate two important properties of neural time-series data: its dynamical nature and its stochasticity. As a student model, they are perhaps the simplest LVM for time-series, yet they are expressive enough to capture real neural dynamics [1] Appendix D shows similar results for linear gaussian models. The HMM has a state space

---

[1] $\mathcal{Q}$ of 0.29 for HMMs vs. 0.24 for GPFA and 0.35 for LFADS

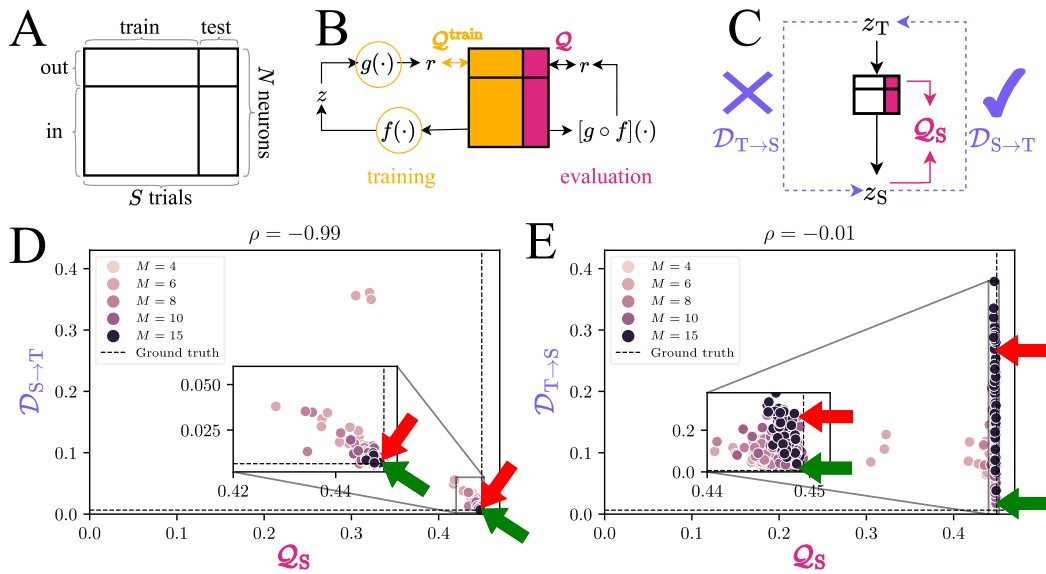

Figure 1: Prediction framework and its relation to ground truth. **A.** To evaluate a neural LVM with co-smoothing, the dataset is partitioned along the neurons and trials axes. **B.** The held-in neurons are used to infer latents $z$, while the held-out serve as targets for evaluation. The encoder $f$ and decoder $g$ are trained jointly to maximise co-smoothing $\mathcal{Q}$. After training, the composite mapping $g \circ f$ is evaluated on the test set. **C.** We hypothesise that models with high co-smoothing may have an asymmetric relationship to the true system, ensuring that model representation contains the ground truth, but not vice-versa. We reveal this in a synthetic student(S)-teacher(T) setting by the unequal performance of regression on the states in the two directions. $\mathcal{D}_{u \to v}$ denote decoding error of model $v$ latents $z_v$ from model $u$ latents $z_u$. **D.** Several student HMMs are trained on a dataset generated by a single teacher HMM. The Student→Teacher decoding error $\mathcal{D}_{S \to T}$ is low and tightly related to the co-smoothing score. **E.** The Teacher→Student decoding error $\mathcal{D}_{T \to S}$ is more varied and uncorrelated to co-smoothing. Dashed lines represent the ground truth, evaluating the teacher itself as a candidate model. A score of $\mathcal{Q} = 0$ corresponds to predicting the mean firing-rate for each neuron at all trials and time points. Green and red arrows represent "Good" and "Bad" models respectively, presented in Fig. 2.

$z \in \{1, 2, \dots, M\}$, and produces observations (emissions in HMM notation) along neurons $\boldsymbol{X}$, with a state transition matrix $\boldsymbol{A}$, emission model $\boldsymbol{B}$ and initial state distribution $\boldsymbol{\pi}$. More explicitly:

$$
\begin{aligned}
A_{m,l} &= p(z_{t+1} = l | z_t = m) &&\forall\, m, l \\
B_{m,n} &= p(x_{n,t} = 1 | z_t = m) &&\forall\, m, n \\
\pi_m &= p(z_0 = m) &&\forall\, m
\end{aligned}
\tag{6}
$$

The same HMM can serve two roles: a) data-generation by sampling from equation 6 and b) inference of the latents from data on a trial-by-trial basis:

$$
\xi_{t,m}^{(i)} = f_m((\boldsymbol{X}_{:,\text{in}})^{(i)}) = p(z_t^{(i)} = m | (\boldsymbol{X}_{:,\text{in}})^{(i)}),
\tag{7}
$$

i.e., *smoothing*, computed exactly with the forward-backward algorithm (Barber, 2012). Note that although $z$ is the latent state of the HMM, we use its posterior probability mass function $\boldsymbol{\xi}_t$ as the relevant intermediate representation. To make predictions of the rates of held-out neurons for co-smoothing we compute:

$$
R_{n,t}^{(i)} = g_n(\boldsymbol{\xi}_t^{(i)}) = \sum_m B_{m,n} \xi_{t,m}^{(i)} \qquad \forall\, n \in \text{out}, 1 \le t \le T, i \in \text{test}
\tag{8}
$$

As a teacher, we constructed a 4-state model of a noisy chain $A_{m,l} \propto \mathbb{I}[l = (m+1) \mod M] + \epsilon$, with $\epsilon = 1e-2$, $\pi = \frac{1}{M}$, and $B_{m,n} \sim \text{Unif}(0,1)$ sampled once and frozen (Fig. 2, left). We

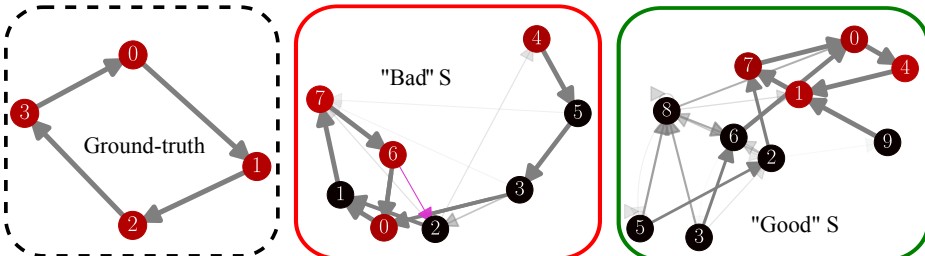

Figure 2: Visualisations of HMMs: the ground truth or teacher model along with two representative extreme student models. Nodes represent states, with colors showing initial state probabilities $\pi_m$ (bright is high probability). Edge width and opacity represents transition probabilities $A_{m,l}$. All three models have high co-smoothing $\mathcal{Q}$ (low $\mathcal{D}_{S\rightarrow T}$). The students differ in $\mathcal{D}_{T\rightarrow S}$ (Fig. 1C,D). Edges with values below 0.02 are removed for visualisation. Note the $(1 \rightarrow 7 \rightarrow 0 \rightarrow 4)$ cycle of the good student, and the $(6 \rightarrow 0 \rightarrow 1 \rightarrow 7)$ cycle in the bad student. They differ in $\pi$, and the latter has an outgoing edge $(6 \rightarrow 2)$, with $A_{6,2} = 0.08$, $A_{6,0} = 0.89$.

generated a dataset of observations from this teacher (see appendix H). We trained $400$ students with $4 - 15$ states on the same teacher data using gradient-based methods (see appendix A). All students had high co-smoothing scores, with some variance, and a trend for large students to perform better. Consistent with our hypothesis, the ability to decode the teacher from the student varied little, and was highly correlated to the co-smoothing score (Fig. 1D). In contrast, the ability to decode the student from the teacher displayed a large variability, and little correlation to the co-smoothing score (Fig. 1E). See appendix B for details of the regression.

What is it about a student model, that produces good co-smoothing with the wrong latents? We consider the HMM transition matrix for the teacher and two exemplar students – named "Good" and "Bad" (marked by green and red arrows in Fig. 1CD) – and visualise their states and transition probabilities using graphs in Fig. 2 . The teacher is a cycle of 4 steps. The good student has such a cycle $(1 \rightarrow 7 \rightarrow 0 \rightarrow 4)$, and the initial distribution $\pi$ is only on that cycle, rendering the other states irrelevant. In contrast, the *bad* student also has this cycle $(6 \rightarrow 0 \rightarrow 1 \rightarrow 7)$, but the $\pi$ distribution is not consistent with the cycle, and there is an outgoing edge from the cycle $(6 \rightarrow 2$, highlighted in pink). Note that this does not interfere with co-smoothing, because the teacher itself is noisy. Thus, occasionally, there will be trials where the teacher will not have an exact period of 4 states. In such trials, the bad model will infer the irrelevant states instead of jumping to another relevant state, as in the teacher model.

## 5 FEW-SHOT PREDICTION SELECTS BETTER MODELS

Because our objective is to obtain latent models that are close to the ground truth, the co-smoothing prediction scores described above are not satisfactory. Can we devise a new prediction score that will be correlated with ground truth similarity? The advantage of prediction benchmarks is that they can be optimized, and serve as a common language for the community as a whole to produce better algorithms (Deng et al., 2009).

We suggest **few-shot co-smoothing** as a complementary prediction score to co-smoothing, to be used on models with good scores on the latter. Similarly to standard co-smoothing, the functions $g$ and $f$ are trained using all trials of the training data (Fig. 3A). The key difference is that a separate group of $N^{k\text{-out}}$ neurons is set aside, and only $k$ trials of these neurons are used to estimate a mapping $g' : \mathbf{Z}_{t,:} \mapsto \mathbf{R}_{t,k\text{-out}}$ (Fig. 3B), similar to $g$ in equation 2. The neural LVM $(f, g, g')$ is then evaluated on both the standard co-smoothing $\mathcal{Q}$ using $g \circ f$ and the few-shot version $\mathcal{Q}^k$ using $g' \circ f$ (Fig. 3C).

This procedure may be repeated several $(s)$ times independently on resampled sets of $k$ trials, giving $s$ estimates of $g'$, each yielding a score $\mathcal{Q}^k$ for each $k$-set. For small $k$, the $\mathcal{Q}^k$s tend to be highly variable. Thus we compute and report the average score $\langle \mathcal{Q}_S^k \rangle$ over the $s$ resamples for each student S. Practical advice on how to choose the value of $k$ and $s$ is given in appendix G.

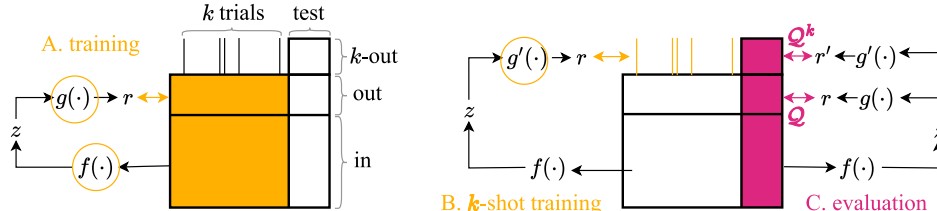

Figure 3: Co-smoothing and few-shot co-smoothing; a composite evaluation framework for Neural LVMs. **A.** The encoder $f$ and decoder $g$ are trained jointly using held-in and held-out neurons. **B.** A separate decoder $g'$ is trained to readout $k$-out neurons using only $k$ trials. Meanwhile, $f$ and $g$ are frozen. **C.** The neural LVM is evaluated on the test set resulting in two scores: co-smoothing $\mathcal{Q}$ and $k$-shot co-smoothing $\mathcal{Q}^k$.

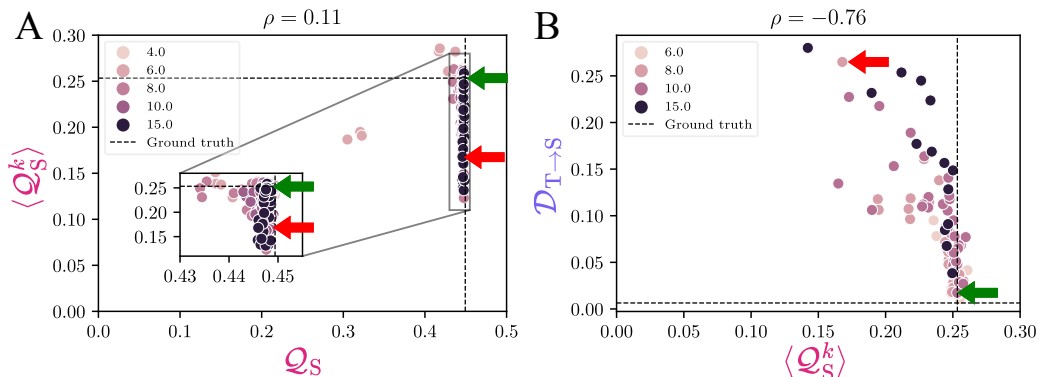

Figure 4: Few-shot prediction selects better models. **A.** Student models with high co-smoothing have highly variable 6-shot co-smoothing and uncorrelated to co-smoothing. **B.** For the set of students with high co-smoothing, i.e., satisfying $\mathcal{Q}_S > \mathcal{Q}_T - 10^{-3}$, 6-shot co-smoothing to held-out neurons is negatively correlated with decoding error from teacher-to-student. Following Fig. 4C,D dashed lines represent the ground truth, green and red arrows represent "Good" and "Bad" models (Fig. 2).

To show the utility of the newly-proposed prediction score, we return to the same HMM students from Fig. 1. For each student, we evaluate $\langle \mathcal{Q}_S^k \rangle$. This involves estimating the bernoulli emission parameters $\hat{B}_{m,k\text{-out}}$, given the latents $\xi_{t,m}^{(i)}$ using equation 11 and then generating rate predictions for the $k$-out neurons using equation 8. First, we see that it provides new information on the models, as it is not correlated with standard co-smoothing (Fig. 4A). We also show that it is not simply a harder version of co-smoothing (appendix C). We are only interested in models that have good co-smoothing, and thus select students satisfying $\mathcal{Q}_S > \mathcal{Q}_T - \epsilon$, choosing $\epsilon = 10^{-3}$. For these students, we see that despite having very similar co-smoothing scores, their $k$-shot scores $\langle \mathcal{Q}_S^k \rangle$ are highly correlated with the ground truth measure $\mathcal{D}_{T\to S}$ (Fig. 4B). Taken together, these results suggest that the combined objective of maximising $\mathcal{Q}_S$ and $\langle \mathcal{Q}_S^k \rangle$ simultaneously – both prediction based objectives – yields models achieving low $\mathcal{D}_{S\to T}$ *and* $\mathcal{D}_{T\to S}$, a more complete notion of model similarity to the ground truth.

## 6  WHY DOES FEW-SHOT WORK?

The example HMM students of Fig. 2 can help us understand why few-shot prediction identifies good models. The students differ in that the *bad* student has more than one state corresponding to the same teacher state. Because these states provide the same output, this feature does not hurt co-smoothing. In the few-shot setting, however, the output of all states needs to be estimated using a limited amount of data. Thus the information from the same amount of observations has to be distributed across more states. This data efficiency argument can be made more precise.

Consider a student-teacher scenario as in section 4. We let $T = 2$ and use a stationary teacher $z_1^{(i)} = z_2^{(i)} = m$. Now consider two examples of inferred students. To ensure a fair comparison, both must have two latent states. In the *good* student, $\xi$, these two states statistically do not depend on time, and therefore it does not have extraneous dynamics. In contrast, the *bad* student, $\mu$, uses one state for the first time step, and the other for the second time step. A particular example of such students is:

$$\xi_t = [0.5 \quad 0.5]^T \quad t \in \{1, 2\} \tag{9}$$

$$\mu_{t=1} = [1 \quad 0]^T \qquad \mu_{t=2} = [0 \quad 1]^T \tag{10}$$

where each vector corresponds to the two states, and we only consider two time steps $t = 1, 2$.

We can now evaluate the maximum likelihood estimator of the emission matrix from $k$ trials for both students. In the case of bernoulli HMMs the maximum likelihood estimate of $g'$ given a fixed $f$ and $k$ trials has a closed form:

$$\hat{B}_{m,n} = \frac{\sum_{i \in k\text{-shot trials}} \sum_{t=1}^{T} \mathbb{I}[X_{t,n}^{(i)} = 1]\xi_{t,m}^{(i)}}{\sum_{i' \in k\text{-shot trials}} \sum_{t'=1}^{T} \xi_{t',m}^{(i')}} \qquad \forall\, 1 \le m \le M \text{ and } n \in k\text{-out neurons} \tag{11}$$

We consider a single neuron, and thus omit $n$. Because both states play the same role, we write the $m = 1$ case:

$$\hat{B}_1(\xi) = \frac{0.5(C_1 + C_2)}{0.5kT} \qquad \hat{B}_1(\mu) = \frac{C_1}{k} \tag{12}$$

where $C_t$ is the number of times $x$ occurs at time $t$ in $k$ trials. We see that $C_t$ is a sum of $k$ i.i.d Bernoulli random variables with the teacher parameter $B^*$, for both $t = 1, 2$.

The expected value of both quantities is the same ($B^*$), but the good student, $\xi$, averages over more Bernoulli samples ($kT$ samples as opposed to $k$ in the bad student, $\mu$), and hence has a smaller variance. We show in appendix **??** that this larger variability translates to lower performance on average. Overall we see that every time that a student has an extra state instead of reusing existing states, this costs the estimator more variance. In appendix K we show a similar argument for continuous state models.

## 7 SOTA LVMS ON NEURAL DATA

In section 4 we showed that models with near perfect co-smoothing may possess latents with extraneous dynamics. We established this in a synthetic student-teacher setting with simple HMM models.

To show the applicability in more realistic scenarios, we trained several models from two SOTA architectures, LFADS (Sedler & Pandarinath, 2023; Pandarinath et al., 2018; Keshtkaran et al., 2022), a variational autoencoder (Kingma, 2013), and STNDT (Le & Shlizerman, 2022; Ye & Pandarinath, 2021), a transformer (Nguyen & Salazar, 2019; Huang et al., 2020), on mc_maze_20 consisting of neural activity recorded from monkeys performing a maze solving task (Churchland et al., 2010), curated by Pei et al. (2021). The 20 indicates that spikes were binned into $20ms$ time bins. We evaluate co-smoothing on a test set of trials and define the set of models with the best co-smoothing (appendix E and H).

An integral part of LFADS and STNDT training is the random hyperparameter sweep which generates several candidate solutions to the optimization problem equation 5.

With each model $f_u$, we infer latents evaluated over a fixed set of test trials $\mathcal{X}^{\text{test}}$, using equation 1.

In the HMM case, we had ground truth that enabled us to directly compare the student latent to that of the teacher. With real neural data we do not have this privilege. To nevertheless reveal the

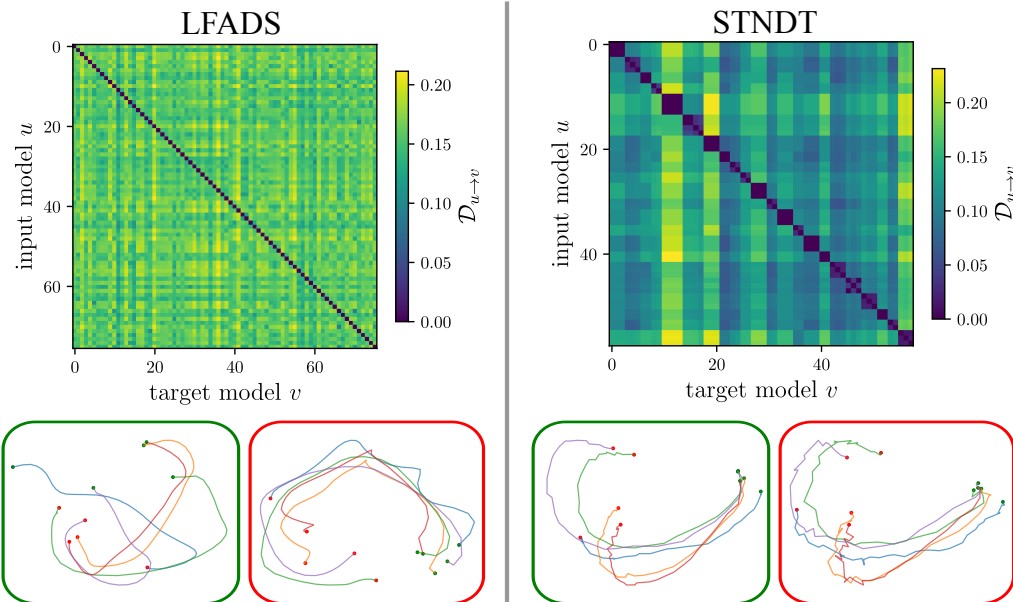

Figure 5: Cross-decoding as a proxy for distance to the ground truth in near-SOTA models. 200 LFADS models (left) and 120 STNDT models (right) were trained on the `mc_maze_20` dataset then selected for high-cosmoothing (appendix E). The latents of each pair of models were decoded from one another, and the decoding error is shown in the matrices. Good models are expected to be decoded from all other models, and hence have low values in their corresponding columns. **Bottom left:** Trajectories of two LFADS models, with the lowest (left and in a green box, best model:= $\operatorname{argmin}_v \langle \mathcal{D}_{u \to v} \rangle_u$) and highest (right and in the red box, the worst model:= $\operatorname{argmax}_v \langle \mathcal{D}_{u \to v} \rangle_u$) column averaged cross-decoding errors, projected onto their leading two principal components. Scores for these models are indicated in Fig. 6 by the arrows. Each trace is the trajectory for a single trial, starting at a green dot and ending at a red dot. **Bottom right** Same for STNDT.

presence or absence of extraneous dynamics, we instead compare the models to each other. The key idea is that all models contain the teacher latent, because they have good co-smoothing. One can then imagine that each student contains a selection of several extraneous features. The best student is the one containing the least such features, which would imply that all other students can decode its latents, while it cannot decode theirs. We therefore use *cross-decoding* among student models as a proxy to the ground truth.

Instead of computing $\mathcal{D}_{S \to T}$ and $\mathcal{D}_{T \to S}$ as in section 4 we perform cross-decoding from latents of model $u$ to model $v$ ($\mathcal{D}_{u \to v}$) for every pair of models $u$ and $v$ using linear regression and evaluating an $R^2$ score for each mapping (appendix E). In Fig. 5 the results are visualised by a $U \times U$ matrix with entries $\mathcal{D}_{u \to v}$ for all pairs of models $u$ and $v$.

We hypothesize that the latents $z_u$ contain the information necessary to output good rate predictions $r$ that match the outputs plus the arbitrary extraneous dynamics. This former component must be shared across all models with high $\mathcal{Q}$, whereas the latter could be unique in each model – or less likely to be consistent in the population. The ideal model $v^*$ would have no extraneous dynamics therefore, all the other models should be able to decode to it with no error, i.e., $\mathcal{D}_{u \to v^*} = 0 \; \forall \; u$. Provided a large and diverse population of models only the 'pure' ground truth would satisfy this condition. To evaluate how close is a model $v$ to the ideal $v^*$ we propose a simple metric: the column average $\langle \mathcal{D}_{u \to v} \rangle_u$. This will serve as proxy for the distance to ground truth, analogous to $\mathcal{D}_{T \to S}$ in Fig. 4. We validate this procedure using the student-teacher HMMs in appendix F, where we show it is highly correlated to ground truth, and as correlated to few-shot as the SOTA models.

Having developed a proxy for the ground truth we can now correlate it with the few-shot regression to held-out neurons. Fig. 6 shows a negative correlation for both architectures, similar to the HMM examples above. As an illustration of the latents of different models, Fig. 5 shows the

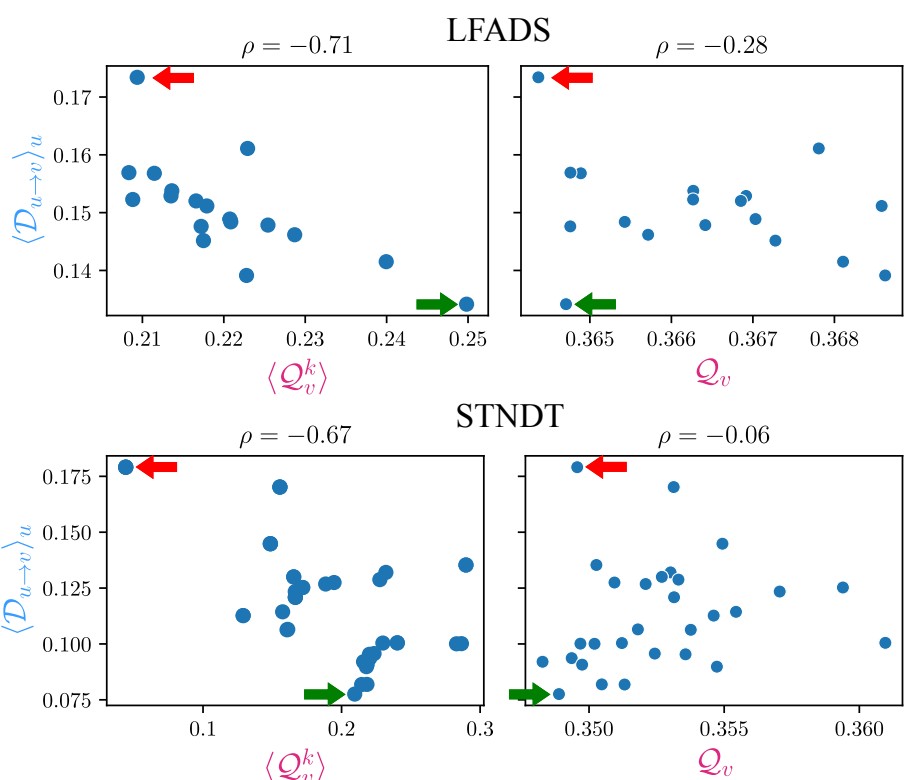

Figure 6: Few-shot scores correlate with the proxy of distance to the ground truth. Several models of two architectures (LFADS top, STNDT bottom) were trained on neural recordings from monkeys performing a maze task, the mc_maze_20 benchmark (Churchland et al., 2010; Pei et al., 2021). Distance to ground truth was approximated by the cross-decoding column average $\langle \mathcal{D}_{u \to v} \rangle_u$ (Fig. 5). Few-shot ($k = 128$) co-smoothing scores (left) negatively correlate with $\mu$, while regular co-smoothing (right) does not. Green and red arrows indicate the extreme models whose latents are visualised in Fig. 5 matched by box/arrow colours. $\mathcal{Q}_v$ values may be compared against an EvalAI leaderboard (Pei et al., 2021). Note that we evaluate using an offline train-test split, not the true test set used for the leaderboard scores, for which held-out neuron data is not directly accessible.

.

PCA projection of several trials from two different models. Both have high co-smoothing scores (LFADS: $0.3647, 0.3643$, STNDT: $0.3488, 0.3495$), but differ in their cross-decoding column average $\langle \mathcal{D}_{u \to v} \rangle_u$. Note the somewhat smoother trajectories in the model with higher few-shot score. It is also possible to cross-decode across the two populations, as shown in Appendix J.

## 8 DISCUSSION

Latent variable models aim to infer the underlying latents using observations of a target system. We showed that co-smoothing, a common prediction measure of the goodness of such models cannot discriminate between certain classes of latents. In particular, extraneous dynamics can be invisible to such a measure.

We suggest a complementary prediction measure: few-shot co-smoothing. Instead of directly regressing from held-in to held-out neurons as is done to evaluate co-smoothing, we distinguish the encoder and the decoder. To evaluate the trained model we substitute the decoder with a new decoder estimated using only a 'few' ($k$) number of trials. The rate predictions provided by the few-shot decoder are evaluated the same as in standard co-smoothing. Using synthetic datasets and HMM models, we show numerically and analytically that this measure correlates with the distance of model latents to the ground truth.

We demonstrate the applicability of this measure to real world neural datasets, with SOTA architectures. This required developing a new proxy to ground truth – cross decoding. For each pair of SOTA models that we obtained, we performed a linear regression across model latents, provided identical input data. Models with extraneous dynamics showed up as a bad target latent on average, and vice versa. Finally we show that these two characterisations of extraneous dynamics are correlated. An interesting extension would be to use this new metric as another method to select good models. The computational cost is high, because it requires training a population of models and comparing between all of them. It is also less universal and standardised than few-shot co-smoothing, as it is dependent on a specific 'jury' of models. The HMM results, however, show that it is more correlated to ground truth than the few-shot.

While the combination of student-teacher and SOTA results put forth a compelling arguement, we address here a few limitations of our work. Firstly, our SOTA results use only one of the datasets in the benchmark suite (Pei et al., 2021). With regard to the few-shot regression, while the bernoulli HMM scenario has a closed form solution: the maximum likelihood estimate, the poisson GLM regression for the SOTA models is optimised iteratively and is sensitive to the l2 hyperparameter `alpha`. In our results we select $k$ and $\alpha$ that distinguish models in our candidate model sets giving moderate/high few-shot scores for some models and low scores to others. This is an empirical choice that must be made for each dataset and model-set. The few-shot training of $g'$ is computationally inexpensive and may be thus can evaluated over a range of values to find the ideal ones.

Overall, our work advances latent dynamics inference in general and prediction frameworks in particular. By exposing a failure mode of standard prediction metrics, we can guide the design of inference algorithms that take this into account. Furthermore, the few-shot prediction can be incorporated into existing benchmarks and help guide the community to build models that are closer to the desired goal of uncovering latent dynamics in the brain.

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

## A  HIDDEN MARKOV MODEL TRAINING

HMMs are traditionally trained with expectation maximisation, but they can also be trained using gradient-based methods. We focus here on the latter as these are used ubiquitously and apply to a wide range of architectures. We use an existing implementation of HMMs with differentiable parameters: dynamax – a library of differentiable state-space models built with jax.

We seek HMM parameters $\theta := (A, B^{[\text{in,out}]}, \pi)$ that minimise the negative log-likelihood loss, $L$ of the held-in and held-out neurons in the train trials:

$$L(\theta; \mathcal{X}_{[\text{in,out}]}^{\text{train}}) = -\log p(\mathcal{X}_{[\text{in,out}]}^{\text{train}}; \theta) \tag{13}$$

$$= \sum_{i \in \text{train}} -\log p\left(\left(X_{1:T, [\text{in,out}]}\right)^{(i)}; \theta\right) \tag{14}$$

To find the minimum we do full-batch gradient descent on $L$, using dynamax together with the Adam optimiser (Kingma & Ba, 2014) .

## B  DECODING ACROSS HMM LATENTS: FITTING AND EVALUATION

Consider two HMMs $u$ and $v$, of sizes $M(u)$ and $M(v)$, both candidate models of a dataset $\mathcal{X}$. Following equation 7, each HMM can be used to infer latents from the data, defining encoder mappings $f^u$ and $f^v$. These map a single trial $i$ of the data $(X_{:,\text{in}})^{(i)} \in \mathcal{X}$ to $(\xi_t^{(i)})_u$ and $(\xi_t^{(i)})_v$.

We now perform a multinomial regression from $(\xi_t^{(i)})_u$ to $(\xi_t^{(i)})_v$.

$$\boldsymbol{p}_t^{(i)} = h\left(\left(\boldsymbol{\xi}_t^{(i)}\right)_u\right) \tag{15}$$

$$h(\xi) = \sigma(W\boldsymbol{\xi} + \boldsymbol{b}) \tag{16}$$

where $W \in \mathbb{R}^{M(v) \times M(u)}$, $\boldsymbol{b} \in \mathbb{R}^{M(v)}$ and $\sigma$ is the softmax. During training we sample states from the target PMFs $(z_t^{(i)})_v \sim (\xi_t^{(i)})_v$ thus arriving at a more well know problem scenario: classification of $M(v)$-classes. We optimize $W$ and $\boldsymbol{b}$ to minimise a cross-entropy loss to the target $(\hat{z}_t^{(i)})_v$ using the `fit()` method of `sklearn.linear_model.LogisticRegression`.

We define decoding error, as the average Kullback-Leibler divergence $D_{KL}$ between target and predicted distributions:

$$\mathcal{D}_{u \to v} := \frac{1}{S^{\text{test}}T} \sum_{i \in \text{test}} \sum_{t=1}^{T} D_{KL}\left(\boldsymbol{p}_t^{(i)}, (\boldsymbol{\xi}_t^{(i)})_v\right) \tag{17}$$

where $D_{KL}$ is implemented with `scipy.special.rel_entr`.

In section 4 and Fig. 1, the data $X$ is sampled from a single teacher HMM, T, and we evaluate $\mathcal{D}_{\text{T} \to \text{S}}$ and $\mathcal{D}_{\text{S} \to \text{T}}$ for each student notated simply as S.

## C  FEW-SHOT CO-SMOOTHING IS NOT SIMPLY HARD CO-SMOOTHING

The few-shot benchmark is a more difficult one than standard co-smoothing. Thus, it might seem that any increase in the difficulty of the benchmark will yield similar results. To show this is not the case, we use standard co-smoothing with fewer held-in neurons (Fig. 7). The score is lower (because it's more difficult), but does not discriminate models.

## D  STUDENT-TEACHER RESULTS IN LINEAR GAUSSIAN STATE SPACE MODELS

We demonstrate that our results are not unique to the HMM setting by simulating another simple scenario: linear gaussian state space models (LGSSM), i.e., Kalman Smoothing.

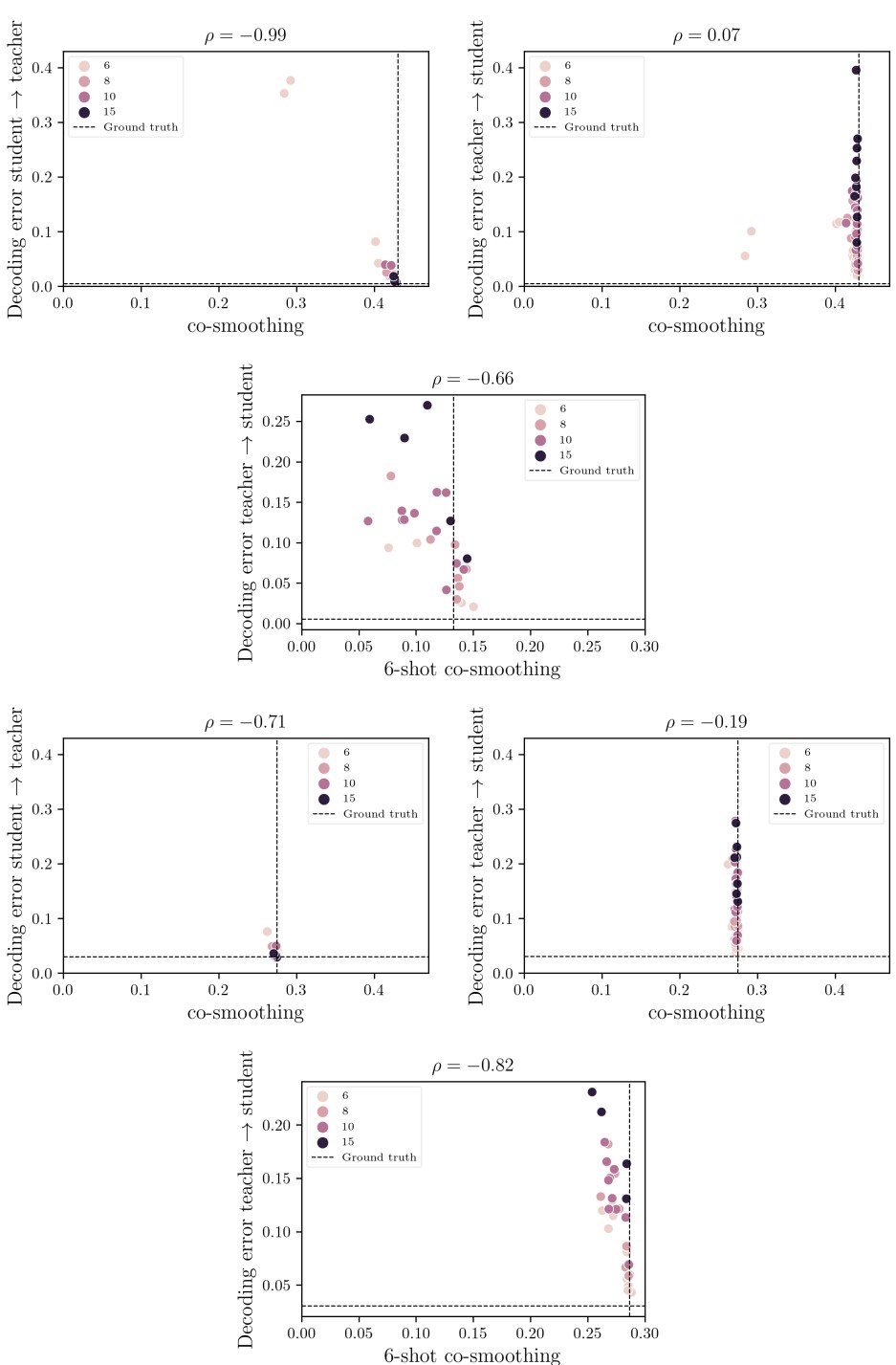

Figure 7: **Making co-smoothing harder does not discriminate between models. Top three:** Increasing the number of held out neurons from $N^{\text{out}} = 50$ to $N^{\text{out}} = 100$. First two panels: Same as main text Fig. 1CD. Lower panel: Same as main text Fig. 4B. **Bottom three:** Decreasing the number of held-in and held-out neurons to $N^{\text{in}} = 5$, $N^{\text{out}} = 5$, $N^{k\text{-out}} = 50$. Panels as in top row. The score does decrease because the problem is harder, but co-smoothing is still not indicative of good models while few-shot is.

The model is defined by by parameters $(\boldsymbol{\mu}_0, \boldsymbol{\Sigma}_0, \boldsymbol{F}, \boldsymbol{G}, \boldsymbol{H}, \boldsymbol{R})$. A major difference to HMMs is that the latent states $\boldsymbol{z} \in \mathbb{R}^M$ are continuous. They follow the dynamics given by:

$$z_0 \sim \mathcal{N}(\boldsymbol{\mu}_0, \boldsymbol{\Sigma}_0) \tag{18}$$

$$z_t \sim \mathcal{N}(\boldsymbol{F}\boldsymbol{z}_{t-1} + \boldsymbol{b}, \boldsymbol{G}) \tag{19}$$

$$x_t \sim \mathcal{N}(\boldsymbol{H}\boldsymbol{z}_t + \boldsymbol{c}, \boldsymbol{R}) \tag{20}$$

Given these dynamics, the latents $\boldsymbol{z}$ can be inferred from observations $\boldsymbol{x}$ using Kalman smoothing, analogous to equation 7. Here we use the jax based dynamax implementation.

As with HMMs we use a teacher LGSSM with $M = 4$, with parameters chosen randomly (using the dynamax defaults) and then fixed. Student LGSSMs are also initialised randomly and optimised with Adam (Kingma & Ba, 2014) to minimise negative loglikelihood on the training data (see appendix H for dimensions of data). $\mathcal{D}_{S \rightarrow T}$ and $\mathcal{D}_{T \rightarrow S}$ is computed with linear regression (`sklearn.linear_model.LinearRegression`) and predictions are evaluated against the target using $R^2$ (`sklearn.metrics.r2_score`). We define $\mathcal{D}_{u \rightarrow v} := 1 - (R^2)_{u \rightarrow v}$. Few-shot regression from $\boldsymbol{z}$ to $\boldsymbol{x}^{k\text{-out}}$ is also performed using linear regression.

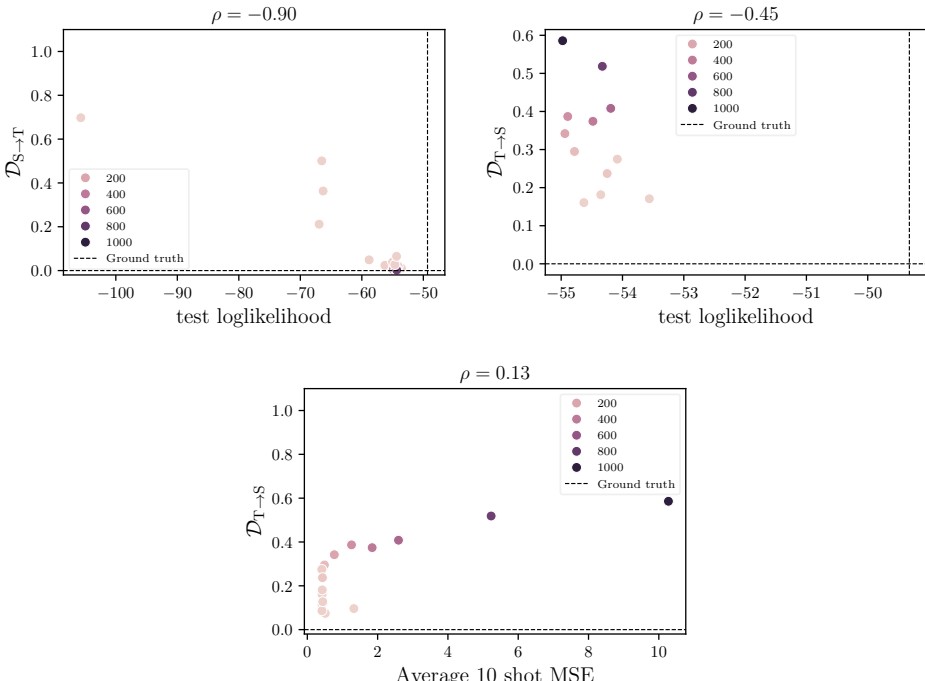

Figure 8: **Left to right**: Student-teacher results for Linear Gaussian State Space Models. We report loglikelihood instead of co-smoothing, and $k$-shot MSE instead of $k$-shot co-smoothing.

# E ANALYSIS OF SOTA MODELS

We denote the set of high co-smoothing models as those satisfying $\mathcal{Q}_{\text{model}} > \mathcal{Q}_{\text{best model}} - \epsilon$, choosing $\epsilon = 5 \times 10^{-3}$ for LFADS and $\epsilon = 1.3 \times 10^{-2}$ for STNDT. $\mathcal{F} := \{(f_u, g_u)\}_{u=1}^{U}$, the encoders and decoders respectively. Note that both architectures are deep neural networks given by the composition $g \circ f$, and the choice of intermediate layer whose activity is deemed the 'latent' $\boldsymbol{Z}$ is arbitrary. Here we consider $g$ the last 'read-out' layer and $f$ to represent all the layers up-to $g$. $g$ takes the form of Poisson Generalised Linear Model (GLM), a natural and simple choice for the few-shot version $g'$. To this end, we use `sklearn.linear_model.PoissonRegressor`. The poisson regressor has a hyperparameter `alpha`, the amount of l2 regularisation. For the results in the main text, $\langle \mathcal{Q}_v^k \rangle$ in Fig. 6, we select $\alpha = 10^{-3}$.

To perform few-shot co-smoothing, we partition the train data into several subsets of $k$ trials. To implement this in a standarised way, we build upon the `nlb_tools` library (appendix I). This way we ensure that all models are trained and tested on identical partitions.

We perform a cross-decoding from the latents of model $u$, $(\boldsymbol{Z}_{t,:})_u$, to those of model $v$, $(\boldsymbol{Z}_{t,:})_v$, for every pair of models $u$ and $v$ using a linear mapping $h(\boldsymbol{z}) := W\boldsymbol{z} + \boldsymbol{b}$ implemented with `sklearn.linear_model.LinearRegression`:

$$\left(\hat{\boldsymbol{Z}}_{t,:}^{(i)}\right)_v = h_{u \to v}\left(\left(\boldsymbol{Z}_{t,:}^{(i)}\right)_u\right) \tag{21}$$

minimising a mean squared error loss. We then evaluate a $R^2$ score (`sklearn.metrics.r2_score`) of the predictions, $(\hat{\boldsymbol{Z}})_v$, and the target, $(\boldsymbol{Z})_v$, for each mapping. We define the decoding error $\mathcal{D}_{u \to v} := 1 - (R^2)_{u \to v}$. The results are accumulated into a $U \times U$ matrix (see Fig. 5).

## F  VALIDATING CROSS-DECODING COLUMN-MEAN AS A PROXY OF GROUND TRUTH DISTANCE IN HMMS

For SOTA models, we don't have ground truth and therefore use cross-decoding as a proxy. We validate this approach in the HMM setting, where we can compute cross-decoding among student models, while also having access to ground truth, i.e., the teacher. As Fig. 9 shows, the novel cross-decoding metric is highly correlated to the ground truth metric of interest $\mathcal{D}_{T \to S}$.

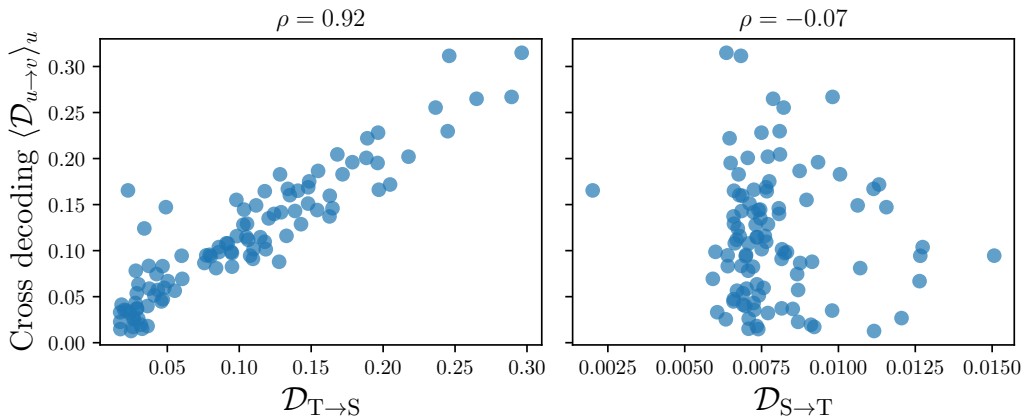

Figure 9: For HMM students with high co-smoothing $\mathcal{Q}_S > \mathcal{Q}_T - 10^{-3}$ (and therefore low $\mathcal{D}_{S \to T}$ 1D), the cross-decoding metric $\langle \mathcal{D}_{u \to v} \rangle_{u \in \text{students}}$ is correlated to ground truth distance $\mathcal{D}_{T \to S}$ and uncorrelated to $\mathcal{D}_{S \to T}$.

Next, in Fig. 10 we replicate the comparison in Fig. 6 of the main text, with the HMMs instead of SOTA models. Despite very different LVM architectures and very different datasets (synthetic versus real neural data), the results are strikingly similar.

Taken together, these results reinforce our use of the novel cross-decoding metric as a proxy to $\mathcal{D}_{T \to S}$ for SOTA models on real data where there is no access to ground truth, i.e., no teacher model T.

## G  HOW TO CHOOSE $k$ AND $s$?

We define $\mathcal{Q}^k$ the $k$-shot co-smoothing score: the co-smoothing score given by predictions from decoder $g'$ trained with only $k$ trials of the $k$-out neurons and the corresponding latents given by the encoder $f$ (section 5 and Fig. 3). As this can be variable across random $k$-trial subsets we report the average $k$-shot co-smoothing, $\langle \mathcal{Q}^k \rangle$, averaging over $s$ decoders each independently trained on

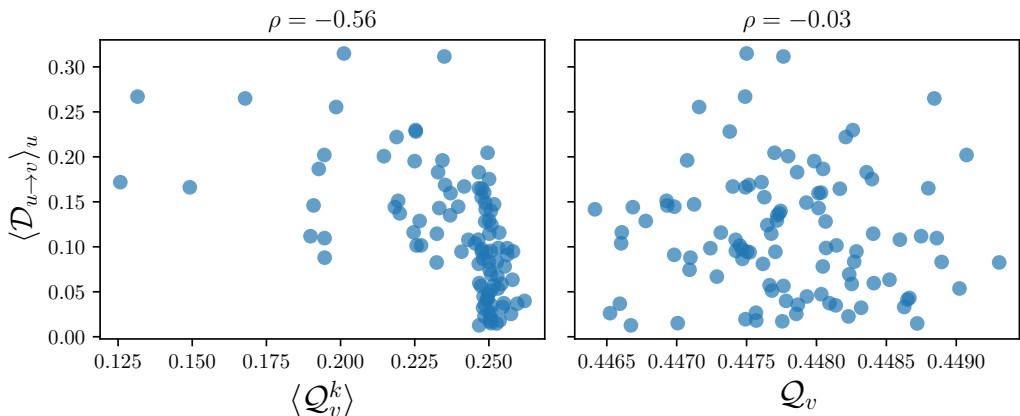

Figure 10: Few-shot co-smoothing validated with cross-decoding in HMMs: a repeat of main text Fig. 6, now in the HMM setting. For HMMs, $v \in$ students, with near-optimal co-smoothing, $\mathcal{Q}_v > \mathcal{Q}_\mathrm{T} - 10^{-3}$, few-shot co-smoothing scores $\langle \mathcal{Q}_v^k \rangle$, with $k = 6$, negatively correlate with the cross-decoding metric $\langle \mathcal{D}_{u \to v} \rangle_u$, used as proxy for the distance from ground truth metric $\mathcal{D}_{\mathrm{T} \to \mathrm{S}}$. Meanwhile, co-smoothing scores $\mathcal{Q}$ are uncorrelated with the same.

random resamples of $k$-trials. Here we report how the results change with $k$, offering guidelines on how to choose $k$ and $s$.

We first analyse the student-teacher HMMs from 4. In Fig. 11 we show several quantities as a function of $k$. We see that small $k$ maximimally separates two extreme models and the scores converge for $k \to \infty$. However at small $k$, scores $\mathcal{Q}^k$ from single models are also more variable, therefore more resamples $s$ are required for a good estimate of the mean $\langle \mathcal{Q}^k \rangle$. We choose $s := \lfloor \frac{S_{\mathrm{train}}}{k} \rfloor$ and find that $k \approx 6$ gives us the best correlation to ground-truth measure (Fig. 11 bottom-right).

We do a similar analysis for LFADS models on the mc_maze_20 dataset. In Fig. 12 we show several values of $Q^k$ (appendix E) for several random samples of $k$-trials, and at various values of $k$. We find that for $k$ values including and below $k = 32$, scores are negative, and at $k = 4$ scores are even worse and vary by orders of magnitude. Among the values we checked, we found $k = 128$ to be the smallest value with positive and low-variance $\mathcal{Q}^k$. Thus, in Fig. 6 we use an intermediate value of $k = 128$ and $s := \lfloor \frac{S_{\mathrm{train}}}{k} \rfloor$.

## H    DIMENSIONS OF DATASETS

We analyse three datasets in this work. Two synthetic datasets generated by an HMM (ground truth in Fig. 2), an LGSMM (appendix D) and the mc_maze_20 dataset from the Neural Latent Benchmarks (NLB) suite (Pei et al., 2021; Churchland et al., 2010). In table 1, we summarise the dimensions of these datsets. To evaluate $k$-shot on the existing SOTA methods while maintaining the NLB evaluations, we conserved the *forward-prediction* aspect. During model training, models output rate predictions for $T^{\mathrm{fp}}$ future time bins in each trial, i.e., equation 1 and equation 2 are evaluated for $1 \leq t \leq T^{\mathrm{fp}}$ while input remains as $\boldsymbol{X}_{1:T,\mathrm{in}}$. Although we do not discuss the forward-prediction metric in our work, we note that the SOTA models receive gradients from this portion of the data.

In mc_maze_20 we reuse held-out neurons as $k$-out neurons. We do this to preserve NLB evaluation metrics on the SOTA models, as opposed to re-partitioning the dataset resulting in different scores from previous works. This way existing co-smoothing scores are preserved and $k$-shot co-smoothing scores can be directly compared to the original co-smoothing scores. The downside is that we are not testing the few-shot on 'novel' neurons. Our numerical results (Fig. 6) show that our concept still applies.

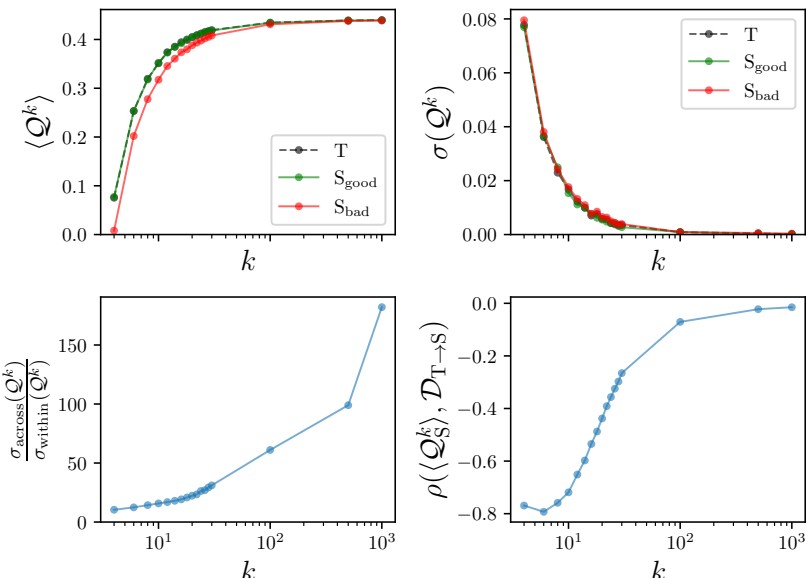

Figure 11: Choosing $k$ and $s$: analysis with HMMs. **Top-left**: Average $k$-shot co-smoothing as a function of $k$ for three models, the teacher T, a good and a bad student as (see 2). **Top-right** Standard deviation of $k$-shot co-smoothing values across resamples. **Bottom-left** *Signal to noise ratio*, ratio of standard deviation of $k$-shot co-smoothing values across models vs with models. **Bottom-right**: Pearson's correlation of average $k$-shot score and the ground-truth decoding measure, for models with high co-smoothing $Q$, as reported in Fig. 4B for $k = 6$. Here we take $s := \lfloor \frac{S_{\text{train}}}{k} \rfloor$.

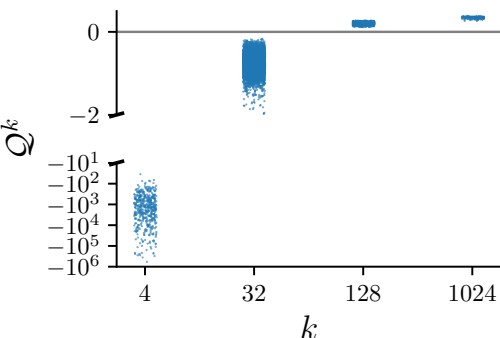

Figure 12: $k$-shot scores (without averaging over $s$ resamples) for LFADS models on the `mc_maze_20` dataset, as a function of $k$. $\mathcal{Q}^k = 0$ is a baseline score obtained by reporting the mean firing rate for each neuron. For small $k$ scores fall below 0 and become highly variable.

Table 1: Dimensions of real and synthetic datasets. Number of train and test trials $S^{\text{train}}$, $S^{\text{test}}$, time-bins per trial for co-smoothing $T$, and forward-prediction $T^{\text{fp}}$, held-in, held-out and $k$-out neurons $N^{\text{in}}$, $N^{\text{out}}$, $N^{k\text{-out}}$.

| Dataset | $S^{\text{train}}$ | $S^{\text{test}}$ | $T$ | $T^{\text{fp}}$ | $N^{\text{in}}$ | $N^{\text{out}}$ | $N^{k\text{-out}}$ |
|---|---|---|---|---|---|---|---|
| Synthetic HMM | 2000 | 100 | 10 | – | 20 | 50 | 50 |
| Synthetic LGSSM (appendix D) | 20 | 500 | 10 | – | 5 | 30 | 30 |
| NLB `mc_maze_20` (Pei et al., 2021; Churchland et al., 2010) | 1721 | 574 | 35 | 10 | 127 | 55 | $55^2$ |

# I    CODE REPOSITORIES

The experiments done in this work are largely based on code repositories from previous works. The code developed here is in `https://osf.io/4bckn/?view_only=73b3aee9a8eb43e8bb3b286c800c6448`. Table 2 provides links to the code repositories used or developed in this work.

Table 2: Summary of key repositories used in this paper

| Repository | Forked from | Citations |
|---|---|---|
| anonymous repo | `https://github.com/neurallatents/nlb_tools` | (Pei et al., 2021) |
| anonymous repo | `https://github.com/trungle93/STNDT` | (Le & Shlizerman, 2022; Ye & Pandarinath, 2021; Pei et al., 2021; Nguyen & Salazar, 2019; Huang et al., 2020) |
| anonymous repo | `https://github.com/arsedler9/lfads-torch` | (Sedler & Pandarinath, 2023; Pandarinath et al., 2018; Keshtkaran et al., 2022) |

# J    COMPATIBILITY AND CONSISTENCY OF CROSS-DECODING ACROSS LVM ARCHITECTURES

In this section we analyse the cross-decoding approach, pooling together the SOTA models from the two architectures: STNDT and LFADS, all trained on the same `mc_maze_20` dataset. We filtered models to those with near-SOTA co-smoothing, specifically $0.348 < \mathcal{Q} < 0.36$, resulting in 75 LFADS and 40 STNDT models. Note that this included LFADS models which were not in the main text Fig. 6.

Fig. 13 shows the cross-decoding matrix $\mathcal{D}_{u \to v}$ for all pairs of models $(u, v)$ in this combined set, as computed in section 7 and appendix E. The cross-decoding matrix reveals a block structure, suggesting larger decoding errors for model pairs from different architectures versus model pairs within the same architecture. Crucially, on top of this block structure, we see clear continuation of columns. This implies that models that are extraneous in one class are also judged as extraneous by the other class. This is summarised in Fig. 14, where we compare column means for each model over the 'same architecture pool' and 'other architecture pool'. Thus, cross decoding can be used across architectures. One should note, however, that having an unbalanced sample from the two classes could bias scores to be lower for the larger class. Finally, we use the combined cross-decoding matrix to repeat the analysis of the main text, but combining both model types. Fig. 15

---

[2]In `mc_maze_20` we use the same set of neurons for $N^{\text{out}}$ and $N^{k\text{-out}}$.

shows that our conclusions hold – co-smoothing is uncorrelated with cross-decoding, while few-shot is correlated.

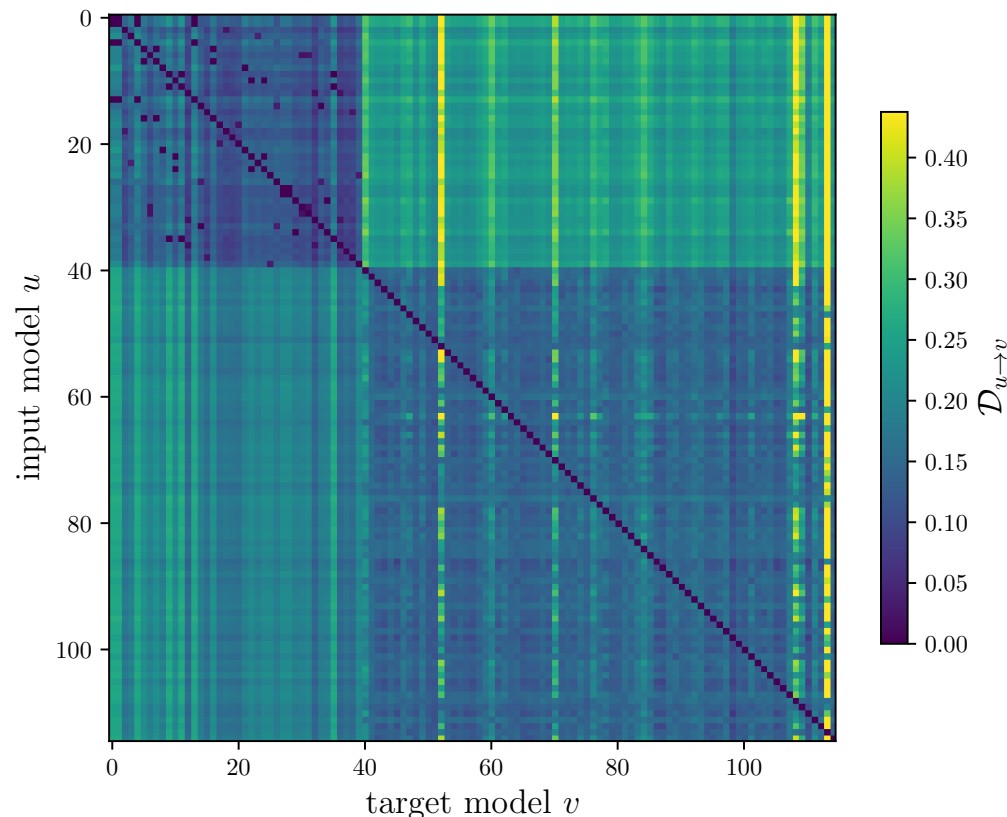

Figure 13: Cross decoding matrix $\mathcal{D}_{u \to v}$ for all model pairs $(u, v)$ from the combined set of 75 LFADS and 40 STNDT models on `mc_maze_20` with near-SOTA co-smoothing $0.345 < \mathcal{Q} < 0.36$. The colormap saturates at the upper 99% quantile of scores in the matrix to better visualise the bulk of the data.

## K  FEW-SHOT ERROR IN CONTINUOUS STATE SPACE

In the main text, we showed that for HMMs, a model with extraneous states gives rise to noisy estimators, and thus to worse few-shot performance. In Appendix D we empirically showed a similar result for a continuous class of models. Here we provide a proof for a simplified setting in the continuous case.

As in the HMM case, we consider two students that can both perfectly predict the observations. One of the students does so in a compact manner, so its $z$ only contains a noisy version of the observations. The other model also has components of its latent $z$ that do not affect the observation. With enough trials, regression will ignore these extraneous directions.

For simplicity, consider where the latent $z \in \mathbb{R}^2$ is a noisy version of the data $x \in \mathbb{R}$. For $k$-shot regression, the data can be described as $\boldsymbol{X} \in \mathbb{R}^{1 \times K}$ and the latents $\boldsymbol{Z} \in \mathbb{R}^{2 \times K}$. More precisely, we formulate the latents as:

$$\boldsymbol{Z} := \boldsymbol{B}\boldsymbol{X} + \boldsymbol{N}, \tag{22}$$

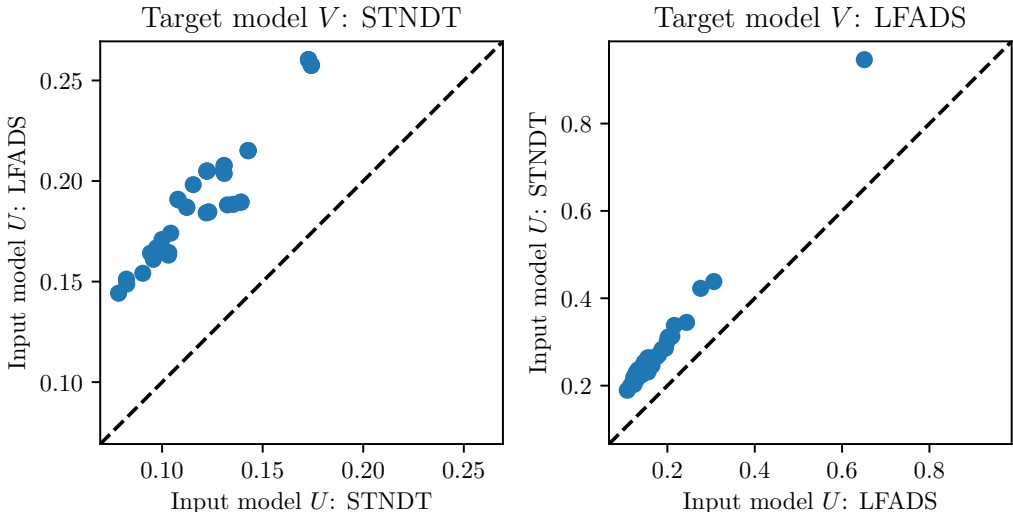

Figure 14: Architecture-wise column means of the cross-decoding matrix. Compare their cross-decoding column means for input models from models of the same architecture versus models of a different architectures, i.e., $\langle \mathcal{D}_{u \rightarrow v} \rangle_{u \in U \setminus \{v\}}$ for target models $v \in V$. LEFT: $V$ is the population of STNDT models and RIGHT: LFADS. All models in $U \cup V$ have near-SOTA co-smoothing, in the range $0.345 < \mathcal{Q} < 0.36$. We do not include the self-decoding scores $\mathcal{D}_{u \rightarrow u}$ as these are trivially near-zero and bias the results.

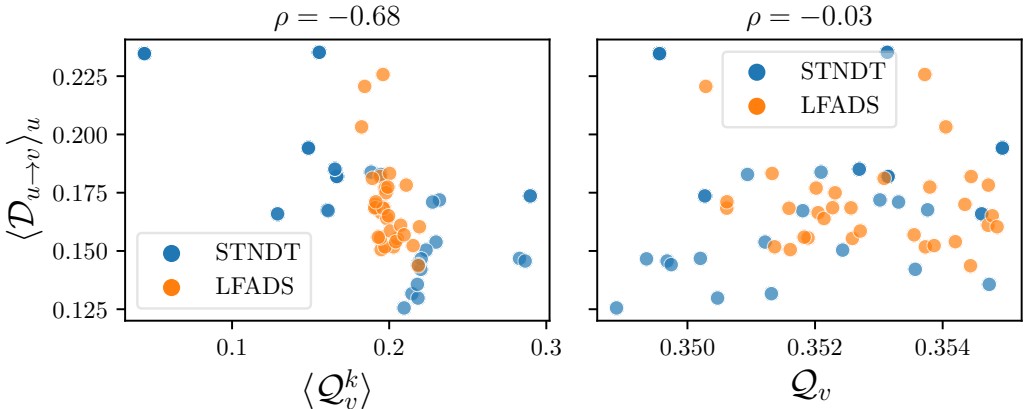

Figure 15: Few-shot scores correlate with the proxy of distance to the ground truth, even in a mixed population of architectures. We repeat the analysis in main text Fig. 6, pooling together the STNDT and LFADS models with high co-smoothing scores $\mathcal{Q} > 0.348$ and compute cross-decoding $\langle \mathcal{D}_{u \rightarrow v} \rangle_u$ for the combined population. To replicate the main text result, we plot models in narrow $\mathcal{Q}$ range, i.e., an upper co-smoothing limit of $\mathcal{Q} < 0.355$, while ensuring that models of both architectures are included.

where $\boldsymbol{B} \in \mathbb{R}^{2 \times 1}$ is an encoding matrix. The two models will differ in their noise term $\boldsymbol{N}$. The compact model has less noise in the directions orthogonal to $\boldsymbol{B}$. This is because extraneous latents imply variability in directions that are not needed for decoding the observations $x$.

For our few-shot regression, we would like to obtain weights $\boldsymbol{a} \in \mathbb{R}^2$, such that $\boldsymbol{a}^T \boldsymbol{z}$ is similar to $x$. The test error is given by:

$$\mathcal{L} = \mathbb{E}_{x,\boldsymbol{z}}(\boldsymbol{a}^T \boldsymbol{z} - x)^2 \tag{23}$$

$$= \mathbb{E}_{x,\boldsymbol{n}} \left( \boldsymbol{a}^T \left( \boldsymbol{B}x + \boldsymbol{n} \right) - x \right)^2 \tag{24}$$

$$= \mathbb{E}_{x,\boldsymbol{n}} \left( \boldsymbol{a}^T \boldsymbol{B}x + \boldsymbol{a}^T \boldsymbol{n} - x \right)^2 \tag{25}$$

$$= \mathbb{E}_{x,\boldsymbol{n}} \left( \left( \boldsymbol{a}^T \boldsymbol{B} - 1 \right) x + \boldsymbol{a}^T \boldsymbol{n} \right)^2 \tag{26}$$

$$= (\boldsymbol{a}^T B - 1)^2 + \mathrm{Tr}[\boldsymbol{a}\boldsymbol{a}^T \Sigma_n] \tag{27}$$

The $a$ obtained by linear regression is given by:

$$\boldsymbol{a} = \boldsymbol{C}_{zz}^{-1} \boldsymbol{C}_{zx}, \tag{28}$$

where $\boldsymbol{C}_{zz} = \boldsymbol{Z}\boldsymbol{Z}^T$ and $\boldsymbol{C}_{zx} = \boldsymbol{Z}\boldsymbol{X}^T$.

For the expected few-shot test error we have:

$$\mathbb{E}_{\boldsymbol{a}}\mathcal{L} = \mathbb{E}_{\boldsymbol{a}}(\boldsymbol{a}^T B - 1)^2 + \mathrm{Tr}[\boldsymbol{a}\boldsymbol{a}^T \Sigma_n] \tag{29}$$

$$= \mathrm{Tr}[\Sigma_{\boldsymbol{a}}\boldsymbol{B}\boldsymbol{B}^T] + \bar{a}^T \boldsymbol{B}\boldsymbol{B}^T \bar{a} + 1 - 2\bar{a}^T \boldsymbol{B} + \mathrm{Tr}\left[\Sigma_{\boldsymbol{a}}\Sigma_n\right] + \bar{a}^T \Sigma_n \bar{a} \tag{30}$$

where $\bar{a}$ and $\Sigma_{\boldsymbol{a}}$ are the mean and covariance of the regression-weight estimates.

For simplicity we choose $\boldsymbol{B} = \begin{bmatrix} 1 & 0 \end{bmatrix}^T$, $x \sim \mathcal{N}(0,1)$, $\boldsymbol{n} \sim \mathcal{N}(0, \Sigma_{\boldsymbol{n}})$, where $\Sigma_{\boldsymbol{n}} = \begin{bmatrix} \sigma_{\mathrm{obs}}^2 & 0 \\ 0 & \sigma_{\mathrm{ext}}^2 \end{bmatrix}$. $\sigma_{\mathrm{obs}}$ is an observation noise that affects the link between the original data $x$ and the estimated readout $\hat{x}$ while $\sigma_{\mathrm{ext}}$ is an extraneous noise orthogonal to the coded variable $x$ in $\boldsymbol{z}$ and corresponds to how extraneous a model is. In this case, the expected few-shot error simplifies to the following:

$$\mathbb{E}_{\boldsymbol{a}}\mathcal{L} = \underbrace{(1 - \bar{a}_1)^2}_{\mathrm{I}} + \underbrace{\mathrm{Var}(a_1)[1 + \sigma_{\mathrm{obs}}^2]}_{\mathrm{II}} + \underbrace{\mathrm{Var}(a_2)\sigma_{\mathrm{ext}}^2}_{\mathrm{III}} + \underbrace{\bar{a}_1^2 \sigma_{\mathrm{obs}}}_{\mathrm{IV}} + \underbrace{\bar{a}_2^2 \sigma_{\mathrm{ext}}^2}_{\mathrm{V}} \tag{31}$$

We also obtain that $\bar{a}_1 = \frac{1}{1+\sigma_{\mathrm{obs}}^2}$ and $\bar{a}_2 = 0$. Thus term III is the only term with significant dependence on $\sigma_{\mathrm{ext}}$. As the model becomes more extraneous, this term grows, and so does the few-shot error. The $\sigma_{\mathrm{ext}}$ dependence is amplified for 'few'-shot, i.e., small $k$, since the $\mathrm{Var}(a_2)$ is larger.

