# OpenReview forum: "When predict can also explain: few-shot prediction to select better neural latents"
_ICLR.cc/2025/Conference — Submitted to ICLR 2025_

### Official Review · Reviewer_UgCb · 2024-10-27

**Soundness:** 3
**Presentation:** 3
**Contribution:** 2
**Rating:** 5
**Confidence:** 2

**Summary:**

This paper focuses on latent variable models (LVMs) for inferring neural activities (neural spikings). Conventional approaches in this field rely on co-smoothing, which jointly estimates the latent variables and predicts observations. This work hypothesizes that, accurate prediction via co-smoothing ensures that LVMs captures the true latent structure, while the reverse may not be true. This claim is justified using an example of discrete-time Hidden Markov Model (HMM). To overcome the aforementioned limitation, this paper proposes an alternative metric called few-shot co-smoothing, which reduces the number of data (trials) used for regression. Experiments on real neural data are conducted with two SOTA methods to validate the effectiveness of the advocated metric.

**Strengths:**

1. The writing of this paper is good, and the illustration figures are clear.

2. The example provided in Section 4 and corresponding observations are interesting.

**Weaknesses:**

Major concerns:

1. The proposed few-shot co-smoothing is based on empirical observations on a simple HMM model. It would be highly beneficial if theoretical results regarding extending this analysis to more generic cases can be provided. In the current manuscript, it is hard to conclude for which family of problems and under what conditions the proposed metric can be effective. It is unknown whether and how few-shot co-smoothing will also help when the relationship between the latent variables and observations becomes complicated.

2. The relationship between the claim, "good prediction guarantees that the true latents are contained within the inferred ones, but not vice versa", and the example provided in Figure 1.DE is unclear. In my understanding, both $\mathcal{D}_{\rm S\rightarrow T}$ and $\mathcal{D} _{\rm T\rightarrow S}$ are utilized to assess whether "the true latents ($z_T$) are contained within the inferred ones ($z_S$)". The paper has not clearly explained their differences, and what each metric indicates. Furthermore, Appendix B leverages multinomial regression to "align" two multinomial distributions and then measure their KL distance. Why is multinomial regression performed with a logistic model rather than other models? It is also possible that two distributions are close under certain mapping but logistic model fail to identify this mapping. To measure the distance between two distributions, there can be some other alternative approaches such as optimal transport. The rationale behind this design choice needs further clarification.

3. Experiments are not sufficiently convincing due to the lack of quantitative comparisons. Section 7 showcases $\mathcal{D}_{\rm S\rightarrow T}$ and $\mathcal{D} _{\rm T\rightarrow S}$ when applying few-shot co-smoothing to SOTA LVMs. However, it remains unknown how mismatched the latents can be with co-smoothing, and how this will affect the prediction performance of the model.

4. No supplementary files or codes are provided, making it hard to verify the reported results.

Minor comments:
1. In line 111, "representing" should be "represents".

2. In line 142, add a comma after "log-likelihood".

3. "Figure 1B" in line 159 should be corrected to "Figure 1C".

4. The cross references for equations are not in correct formats; "\eqref" should be used instead of "\ref".

5. The abbreviations for "Figure" are not consistent throughout the paper. For instance, lines 234, 241, and 244 respectively adopts "fig.", "Fig.", and "FIG".

**Questions:**

See above.

---

> ### Author Response · Authors · 2024-11-18
> **Addressing Scope and Cross-Decoding Clarifications**
>
> We thank the reviewer for the detailed comments. We address the concerns below. The proposed changes will be shared in the updated pdf in coming days.
>
> ### Response to Major Concerns:
>
> 1. **Scope of applicability.**
> 	1. We showed empirical results for HMM, Linear Gaussian State space models (Appendix E), and SOTA. Indeed, we only showed theoretical results for the HMM. We think, however, that the theoretical results are more general. The estimator variance argument we made in the paper actually does not rely on the HMM architecture. We only assume that on two different time points there are same (vs. different) latents with the same observation. If the latents are discrete, as in HMMs (but not limited to HMM), we show how the variance grows. If the latents are continuous (as in the LGSSM in appendix E), we believe it is possible to show a similar theoretical result - and will update the reviewers once we have it.
>
> 	2. **Complex relation from latent to observations**: It is true that in our work we implicitly assume a ‘simple’ relation $g$ between the true latents and observables. If $g$ was more complex, e.g a deep neural network we can redefine $g$ to be just the last layer, absorbing the remaining layers into $f$, arriving at a new latent with a simpler relationship to the observations. Thus, we are free to choose the latent as the last layer, from which it is possible to linearly decode the observables (Similar to Sorscher et al 2022).
> 2. **The idea of latents 'contained' within one another:** We apologise for the lack of clarity. We noticed that the definition of $\mathcal D_{\text{S}\rightarrow \text{T}}$ came too late in the paper, and there were also places where the wrong panel was referenced from the text. These will be amended in the revision.  $\mathcal D_{\text{S}\rightarrow \text{T}}$ represents how well teacher states can be decoded from student states,  i.e a regression is performed from student states to teacher states and the test error as evaluated in Appendix B is defined as $\mathcal D_{\text{S}\rightarrow \text{T}}$. $\mathcal D_{\text{T}\rightarrow \text{S}}$ differs in that regression is performed from teacher states to student states. Thus low $\mathcal D_{\text{S}\rightarrow \text{T}}$ implies student latents contain the teacher latents and low $\mathcal D_{\text{T}\rightarrow \text{S}}$ implies the reverse, that teacher latents contain the student.
> 3. **Clarification:** There appears to be a misunderstanding regarding Section 7. To clarify, our experiments in Section 7 focus on real neural recordings, where there is no student-teacher setup (no S and T). Instead, we train multiple models on the same data and evaluate their pairwise similarity using $\mathcal D_{\text{u}\rightarrow\text{v}}$ metrics, as described. If we understand the comment correctly, "how mismatched the latents can be" is exactly what we are trying to approximate using cross decoding, because we don't have access to the true latents in the neural recording scenario. "How this will affect prediction performance" -- assuming prediction means co-smoothing, we only select models with good performance. Our focus is on how different latents can give rise to equal performance.
> 3. **Code**: Code is available in Appendix J titled ‘Code’ and clicking ‘anonymous repo’. For some reason, this hyperlink was not clearly marked in the PDF in some viewers. It can also be [accessed here](https://osf.io/4bckn/?view_only=73b3aee9a8eb43e8bb3b286c800c6448). We apologise as it was not referenced in the main text. We will fix this.
>
> ### Response to Minor Concerns:
>
> We thank the reviewer for the detailed list of corrections. We will make the corrections and proofread for more such cases.
>
> Regarding point 3: we verified that we already use \eqref for all equations and \ref for figures. With the ICLR style file it appears as ‘equation 3’, the word ‘equation’ is not in our latex script
>
> ### References
> Sorscher, Ben, Surya Ganguli, and Haim Sompolinsky. "Neural representational geometry underlies few-shot concept learning." *Proceedings of the National Academy of Sciences* 119.43 (2022): e2200800119.

---

> > ### Comment · Reviewer_UgCb · 2024-11-24
> >
> > Thanks for the response, which addresses most of my concerns. I will therefore increase my score to 5, yet still with a low confidence score.

---

> > > ### Author Response · Authors · 2024-11-28
> > > **Scope of applicability, theoretical analysis**
> > >
> > > Dear reviewer,
> > >
> > > Thank you for your update.
> > >
> > > To address the concern about scope of applicability and generality of few-shot, we did a theoretical analysis with 2D linear regression, in a continuous latent space setting. It shows how extraneous noise lowers average few-shot score. This is available in the latest pdf in appendix K. If this clarifies some of your concerns, please do consider raising your score further, and let us know if you have any further questions.
> > >
> > > best,
> > > authors

---

### Official Review · Reviewer_RGUv · 2024-11-01

**Soundness:** 3
**Presentation:** 4
**Contribution:** 3
**Rating:** 6
**Confidence:** 3

**Summary:**

The paper studies the limitations of the commonly used co-smoothing prediction framework applied to latent variable models predicting underlying dynamics, specifically in the context of neural activity modeling. The authors provide synthetic teacher-student setup demonstrating how good performance in terms of co-smoothing does not imply learning the ground-truth latents. In the same setup, an additional prediction score termed few-shot co-smoothing is suggested leading to better correlation with true latents for models with high co-smoothing. In a real-world setup, the authors suggest a proxy metric assessing the similarity of a model’s latents to an unknown teacher, and show that their suggested score correlates with it.

I am relatively unfamiliar with the subject of the paper and prior work but found the presentation and ideas easy to follow.

**Strengths:**

* The paper is well organized and each section has a clear and consice statement.

* The teacher-student setup provides a convincing example of the problem with co-smoothing scores.

* The suggested solution of few-shot co-smoothing is simple and seems to be reasonably effective in the suggested setups.

**Weaknesses:**

* Evaluations in section 7 rely on a metric, cross-decoding error, defined by the authors themselves and argued for in the text. No prior work pointing to the validity of the score is provided, see further questions below.

* Since correlation in section 7 is not perfect, and due to the above, further empirical verification of the method, with either additional datasets and/or models, is missing.

**Questions:**

* My main questions revolve around evaluations in section 7 and the cross-decoding metric used as ground truth:
   * Can the authors provide the performance of the cross-decoding score in the synthetic setup where the latent variables are known? can one use it to choose a model with latent representation closest to the ground truth?
   * In the general case and assuming no computational constraints, is this metric supposed to be superior to scores from few-shot cosmoothing and can this be verified empirically?
   * Even if the assumption on cross-decoding as a proxy to the ground truth is correct - selecting from a pool of models that differ by hyperparameters can be highly dependent. Is it possible to select jointly from a pool of different architectures? .e.g, LFADS, STDNT.

* In section 3 can you clarify which notation (bold/regular) refers to vectors/tensors/scalars? I could not understand from the text.

---

> ### Author Response · Authors · 2024-11-18
> **Addressing Cross-Decoding Validation, Novel Metric Contributions, and Notational Consistency**
>
> We thank the reviewer for the detailed comments. We address the concerns below. The proposed changes will be shared in the updated pdf in coming days.
>
> ### Response to Weaknesses:
>
> **Cross decoding:** Indeed, our SOTA results used a novel metric for evaluating model extraneousness - cross decoding. We feel that this novelty is a strength of our contribution: a useful tool to characterise the space of model solutions when there is no ground truth.
>
> Because it is a novel metric, it was vital to validate it on a case where we have ground truth - which was done in Appendix G (and mentioned in line 427). There, we show a very strong correlation between the new measure and the ground truth on HMMs. We will also add the relation between cross-decoding and few-shot on HMMs.
>
> Furthermore, while our metric is novel, it is also related to previous suggestions in the literature. A recent preprint (Versteeg et al. 2023) addresses the same phenomenon of extraneous dynamics for neural LVMs on synthetic data, and defines a measure called Cycle consistency. This is the performance of a decoder from the log of predicted rates $\log r_t$ to $z_t$. This is not exactly our measure of $\mathcal D_{\text{T}\rightarrow \text{S}}$, but has similarities to it. Note that they also evaluate $\mathcal D_{\text{T}\rightarrow \text{S}}$ and refer to it as 'state $R^2$'.
> ### Response to Questions:
> 1. **Questions about cross-decoding:**
> 	1. **Cross-decoding in synthetic setup.** We have done precisely this in Appendix G. We evaluated pair-wise cross-decoding among the HMM students and compared the novel metric to the ground-truth metric. We find a strong correlation.
>
> 	2. **Cross-decoding as a superior score.** This is an interesting idea that we have not considered. The computational cost can be quite prohibitive in general, but it is an interesting suggestion nevertheless. Note that cross-decoding is qualitatively different from few-shot. It is a property of a population of models - in some sense, it is an ordering within this population. As a benchmark, this means comparing any new model to all existing models, and requiring access to their latents. Meanwhile the few-shot co-smoothing is computed for each model independently. Furthermore, its close relation to co-smoothing (the current standard for benchmarking neural LVMs) makes it easier to incorporate in existing  benchmark pipelines. In our HMM scenario, if we compare figures 4 and 9, it does seem that cross decoding is more correlated to ground truth than few-shot. We will discuss this in the revision. Thanks for the suggestion.
>
> 2. **Notation.** We apologise for the lack of consistency. We will update the notation following the [https://github.com/goodfeli/dlbook_notation](https://github.com/goodfeli/dlbook_notation), provided in the ICLR template such that vectors, matrices, and tensors appear distinctly.
>
> ### References
> 1. Versteeg, Christopher, et al. "Expressive dynamics models with nonlinear injective readouts enable reliable recovery of latent features from neural activity." ArXiv (2023).
> 2. Keshtkaran, Mohammad Reza, et al. "A large-scale neural network training framework for generalized estimation of single-trial population dynamics." _Nature Methods_ 19.12 (2022): 1572-1577.

---

> > ### Comment · Reviewer_RGUv · 2024-11-27
> >
> > Thanks you for the responses and addressing my concerns here and in the revision, I have updated my score.

---

### Official Review · Reviewer_PWjg · 2024-11-03

**Soundness:** 2
**Presentation:** 2
**Contribution:** 2
**Rating:** 6
**Confidence:** 2

**Summary:**

The paper focuses on the evaluation method of neural latent-variable models (LVM). Neural LVM aims to learn representations of neural dynamics in a latent space. This space should be able to represent the ground truth hidden representation space. However, due to the lack of ground truth latent in neural signal data, people propose to evaluate the quality of learned latent via co-smoothing, which jointly estimates latent variables and predicts observations along held-out channels as evaluation metrics. With HMM synthetic data, the paper empirically shows that good co-smoothing performance does not ensure good latent, since the model can learn redundant information to have good prediction results. Therefore, the paper instead proposes the few-shot co-smoothing, which better separates good LVM from bad LVM. The paper further provides mathematical insights based on the prediction variance of co-smoothing and few-shot co-smoothing methods. Lastly, the paper adapts co-smoothing with cross-decoding to real data and shows good "accuracy".

**Strengths:**

1. The paper has a high-level storyline that is easy to follow: from the limitation of co-smooth, proposing new few-shot co-smoothing, reasons for few-shot co-smoothing to work, and how to adapt few-shot co-smoothing to real neural signals.
2. The paper covers both empirical analysis of existing methods, practical improvements based on the existing method, and mathematical interpretations of the proposed improvement.
3. The paper provides an honest discussion of limitations.

**Weaknesses:**

1. Writing of limited works is a bit messy: too many paragraphs, mixing related works with the introduction of the work's methodology and contribution. It would be clearer to provide clean related works as a background and only include a short emphasis on the uniqueness of this paper.

2. The writing in other parts is also a bit confusing and the paper seems to be finished in a rush, missing ","s in several positions: L118 after "forward-prediction", L122 after "During evaluation", L125 after "full time-window", L142 after "log-likelihood", etc, ... Let me know if I misunderstood these parts.

3. The paper lacks an explanation for choosing HHM as a synthetic data model to approximate real neural signals.

4. The paper lacks clarity and technical details, see Questions parts.

**Questions:**

1. L58: LVM is introduced without explanation, and the explanation is finally given in L113.

2. L118: "Data may be held-out in time, e.g forward-prediction or in space, co-smoothing (Pei et al., 2021)." Here, is "held-out in time" the same as "forward-prediction" of signal in the latent space or signal space? Besides, is "Data held-out in space" the same as "co-smoothing"? But what does "Data held-out in space" exactly mean?

3. L149: why is equation (5) "without direct access of $X_{\text {test }}^{\text {out }}$"? From the definition in (3) and (4), I feel $X_{\text {test }}^{\text {out }}$ is required.

4. In section 4, how to validate that the specific setting of HMM is a suitable synthetic data model that can reveal features of real neural signals?

5. In section 5, how to choose "k"? What about selecting "k" for real data, is there a principled rule?

I am willing to hear from authors and other reviewers in addressing my questions and concerns. I will change the score if others' comments solve my questions and convince me about the quality and importance of this work.

---

> ### Author Response · Authors · 2024-11-18
> **Clarifying Methodology, Terminology, and Justifications for Synthetic Data Choices**
>
> We thank the reviewer for the detailed comments. We address the concerns below. The proposed changes will be shared in the updated pdf in coming days.
>
> ### Response to Weaknesses:
>
> 1. **Related work section:** We acknowledge this mix-up. We will clean-up the Related Work section, removing parts about the uniqueness of our work.
> 2. **Typos:** We are sorry for these grammatical errors, and thank you for pointing them out. We will thoroughly proofread the paper before revision.
> 3. **HMM justification:** We agree and provide a brief explanation here. Real neural data is trajectories across time, and therefore a teacher model should be a dynamical system. HMMs were chosen as a tractable dynamical system to show our claims. We highlight specific properties of HMMs that are relevant to our use:
>     - HMMs are perhaps the simplest stochastic dynamical system,
>     - HMMs are expressive enough to capture neural dynamics. Indeed, HMMs appear on the leaderboard on the `mc_maze_20`[(see here)](https://eval.ai/web/challenges/challenge-page/1256/leaderboard/3183). While not the leading models, they outperform other baselines.
>     - They may be used for both data generation and inference of latents. (see line 206) The student and teacher come from the same class of models, allowing direct comparison.
> 4. **Clarity:** See response to Questions.
>
> ### Response to Questions:
>
> 1. **LVM definition:** Thank you for spotting this. LVMs were introduced without the math in lines 42-44, but we forgot to define the acronym there. We believe the position of the mathematical description of LVM (Section 3) can remain unchanged.
> 2. **Held-out definitions:** When rereading this, we agree that the definitions were very dense and not clear enough. These will be made clearer in the revision: "Data may be held-out in time, e.g predicting future data points from the past, or in space, e.g predicting neural activities of one set of neurons (or channels) based on those of another set. The latter is called co-smoothing (Pei et al., 2021)."
> 3. **Equation 5.** Thanks for spotting this typo! It should have been $X_\text{train}$, $R_\text{train}$. Models can access $X_\text{train}$ but not $X_\text{test}$. We hope this clarifies the confusion.
> 4. **HMMs:** See 3. in Response to Weaknesses.
> 5. **Selecting $K$:** We dedicated Appendix H to this topic, where we addressed both synthetic and real data. Indeed, we did not summarize the operative conclusions from this appendix - namely, that $K=128$ is the smallest number for which we have scores above zero and with low variability.

---

> > ### Comment · Reviewer_PWjg · 2024-11-26
> > **Thanks for the feedback**
> >
> > Dear authors,
> >
> > I still don't understand equations 3-5, where should be $X_{train}$ and where should be $X_{test}$, could you please clarify further? There's no updates in the pdf.
> >
> > Best Regards,
> > Reviewer

---

> > > ### Author Response · Authors · 2024-11-26
> > > **clarifying eq 3-5, math notations, and pdf update**
> > >
> > > Dear reviewer,
> > >
> > > Thank you for following up. While we are continuing to incorporate feedback from other reviewers into the draft, we have already incorporated most of your feedback, thus we updated the pdf now.
> > >
> > > Please find in the new pdf, equation 5 corrected along with some clarifications. Note that we made changes to the notations to differentiate scalars/matrices/tensors following the feedback from reviewer RGUv.
> > >
> > > We hope this provides more clarity. We also cleaned up the Related works section.
> > >
> > > Best,
> > > authors

---

> > > > ### Comment · Reviewer_PWjg · 2024-12-01
> > > > **Thanks**
> > > >
> > > > Dear authors, thanks for the clarification. My current questions are addressed. I have increased the score to 6 with lower confidence.

---

### Author Response · Authors · 2024-11-18
**General response addressing reviews**

We thank all the reviewers for noting the importance of the problem we address and the clarity of our presentation. Below, we provide detailed responses to all comments and questions, and we will upload an updated PDF in the coming days.

Here, we highlight two major points raised in the reviews:

1. **Validation of cross-decoding:** Validation in a synthetic setting with access to ground truth is essential and has been conducted, as detailed in Appendix G.
2. **Scope of few-shot co-smoothing:** While inspired by empirical observations on HMMs, few-shot co-smoothing has been validated empirically on linear gaussian state-space models and SOTA models, and analytically on HMMs.

Thank you again for your thoughtful feedback, which has helped us improve our submission.

---

### Author Response · Authors · 2024-11-28
**Updated PDF**

Dear Reviewers,

Thanks again for all the work you put into reviewing our paper, and for the suggestions that helped us improve it.
We uploaded an updated PDF that we believe answers all questions and weaknesses raised in the reviews.
The main changes are:

- Following the suggestion by RGUv, we added a discussion on using cross-decoding as a score. (lines 509-512)
- A proof that few-shot selects better models also in continuous state space (Appendix K)
- We added the correlation between few-shot and cross-decoding to the HMM validation (Appendix F)
- We clarified the hypothesis and description of figure 1 (lines 144-160)
- Rewriting the Related Work section (lines 68-97)
- We added combined cross-decoding for both SOTA models (Appendix J)
- Notations, equation and figure references - all standardized and fixed
- Many clarifications - better pointer to appendices, correcting typos, code links, etc.

Once again - thanks for your input, and please consider updating your score and let us know if there are any remaining questions.
Best,
Authors

---

### Meta-Review · Area_Chair_7W7t · 2024-12-15

**Metareview:**

The paper critiques the commonly used co-smoothing evaluation metric for latent variable models (LVMs), demonstrating its failure to reliably infer true latent dynamics due to the inclusion of extraneous dynamics. The authors propose a new metric, few-shot co-smoothing, which evaluates performance over limited data trials, improving the selection of models with minimal extraneous dynamics. They validate this metric using a teacher-student setup with Hidden Markov Models (HMMs) and extend it to real-world neural data using cross-decoding to assess latent space similarity. Strengths include the clear identification of a critical issue in LVM evaluation, the introduction of an effective and simple metric, and validation through both theoretical analysis and empirical data. However, the work lacks sufficient theoretical generalization beyond HMMs, robust real-world validation, and comparisons to alternative evaluation methods. Additionally, the limited scope and unclear applicability of the few-shot metric in broader LVM settings raise concerns about its utility. While the paper provides interesting insights, these limitations undermine its contributions, leading to a borderline rejection recommendation with encouragement for further development and more comprehensive validation.

**Additional Comments On Reviewer Discussion:**

During the rebuttal period, reviewers raised concerns about the generalizability of the few-shot co-smoothing metric beyond HMMs, its theoretical underpinnings for broader LVMs, the limited exploration of real-world datasets, and clarity issues in notations and explanations. The authors addressed these by expanding theoretical analysis to linear regression in continuous latent spaces, clarifying notations, and adding discussions on metric applicability and experimental limitations. They also revised the manuscript for better organization and readability. Despite these efforts, key concerns about the generalizability of the metric, the scope of real-world validations, and the lack of comparisons to alternative evaluation methods remained unresolved. These unresolved issues weighed heavily in the decision, leading to a borderline rejection, with recognition of the paper’s potential for future work.

---

### Decision · Program_Chairs · 2025-01-22

Reject